# Warming accelerates global drought severity

Solomon H. Gebrechorkos[1,2 ✉], Justin Sheffield[2], Sergio M. Vicente-Serrano[3], Chris Funk[4], Diego G. Miralles[5], Jian Peng[6,7], Ellen Dyer[1], Joshua Talib[8], Hylke E. Beck[9], Michael B. Singer[10,11,12] & Simon J. Dadson[1,8]

Drought is one of the most common and complex natural hazards affecting the environment, economies and populations globally[1–4]. However, there are significant uncertainties in global drought trends[4–6], and a limited understanding of the extent to which a key driver, atmospheric evaporative demand (AED), impacts the recent evolution of the magnitude, frequency, duration and areal extent of droughts. Here, by developing an ensemble of high-resolution global drought datasets for 1901–2022, we find an increasing trend in drought severity worldwide. Our findings suggest that AED has increased drought severity by an average of 40% globally. Not only are typically dry regions becoming drier but also wet areas are experiencing drying trends. During the past 5 years (2018–2022), the areas in drought have expanded by 74% on average compared with 1981–2017, with AED contributing to 58% of this increase. The year 2022 was record-breaking, with 30% of the global land area affected by moderate and extreme droughts, 42% of which was attributed to increased AED. Our findings indicate that AED has an increasingly important role in driving severe droughts and that this tendency will likely continue under future warming scenarios.

Water availability has a critical role in shaping ecosystems, economic activities and human livelihoods. Water is an essential resource for agriculture, energy, industry and domestic use, influencing the overall sustainability and development of societies[7,8]. Droughts are also detrimental for vegetation, reducing the carbon uptake of ecosystems, causing widespread plant mortality[9–11] and leading to significant disruptions in ecosystem functioning and biodiversity loss[12]. They also negatively affect the productivity of annual and perennial crops, exacerbating food insecurity and economic instability[11]. With climate change, there is an expectation that droughts will be more frequent and intense[13], with increased impacts on agricultural, environmental and hydrological systems[14,15]. Observational evidence indicates an increase in hydrological and agricultural drought severity in several regions over recent decades, owing to the widespread increase in atmospheric evaporative demand (AED) as well as regional declines in precipitation[16,17]. Future projections from climate models also suggest a heightened severity of droughts in some regions owing to decreases in precipitation and enhanced AED[18].

Although numerous studies have focused on estimating drought trends and their drivers at the global scale, they have been limited by the quality of available global data[3,4,17,19], which adds uncertainties in the assessment of these trends. Crucially, the extent of the effect of increased AED on drought severity as a consequence of global warming remains inadequately explored[20]. AED intensifies water deficits by enhancing evaporation[11], particularly under low-soil-moisture conditions. Moreover, land–atmosphere interactions can lead to positive feedback whereby drying soils and plants decrease latent heat fluxes, leading to increases in temperature and AED, and further increasing

drought severity[13,21]. Although drought can be characterized in many ways to reflect different meteorological, hydrological and ecological drivers, consideration of the influence of AED with respect to precipitation is crucial to understand how climate change is impacting on changes in drought. Some studies have suggested that AED-based drought metrics may overestimate severity compared with hydrological and ecological indicators[22]. However, this mainly stems from uncertainties in Earth system model projections and the physiological effects of atmospheric carbon dioxide on evaporation[17,23]. Methodological challenges also affect comparisons between drought metrics, but applying consistent statistical approaches shows stronger agreement between AED-inclusive indices[24]. Increasing evidence highlights the role of AED in amplifying ecological drought severity through evaporation[25]. Given the recent rise and projected increase of AED owing to anthropogenic warming[17,18], assessing its contribution to drought severity is essential for adaptation planning.

Nevertheless, previous studies have highlighted significant uncertainties in global-scale drought assessments and in the determination of the role of AED on drought severity, largely owing to the choice of models for AED and meteorological forcing dataset[3,4,20,26]. Thus, in previous studies, the selection of methods and datasets have resulted in conflicting results in global drought patterns[4,5,20], highlighting the need for further research to reduce uncertainties induced by varying methods and forcing datasets. For example, simpler temperature-based methods overestimate AED in humid regions, whereas more comprehensive models such as Penman–Monteith, which consider both radiative and thermodynamic terms, offer more accurate results across different climates and seasons[27,28]. Also, reliable and accurate observations of

[1]School of Geography and the Environment, University of Oxford, Oxford, UK. [2]School of Geography and Environmental Science, University of Southampton, Southampton, UK. [3]Instituto Pirenaico de Ecología, Consejo Superior de Investigaciones Científicas (IPE-CSIC), Zaragoza, Spain. [4]Santa Barbara Climate Hazards Center, University of California, Santa Barbara, CA, USA. [5]Hydro-Climatic Extremes Lab (H-CEL), Ghent University, Ghent, Belgium. [6]Department of Remote Sensing, Helmholtz Centre for Environmental Research - UFZ, Leipzig, Germany. [7]Institute for Earth System Science and Remote Sensing, Leipzig University, Leipzig, Germany. [8]UK Centre for Ecology and Hydrology, Wallingford, UK. [9]Climate and Livability Initiative, King Abdullah University of Science and Technology, Thuwal, Saudi Arabia. [10]School of Earth and Environmental Sciences, Cardiff University, Cardiff, UK. [11]Earth Research Institute, University of California, Santa Barbara, CA, USA. [12]Water Research Institute, Cardiff University, Cardiff, UK. ✉e-mail: solomon.gebrechorkos@ouce.ox.ac.uk

precipitation are crucial for realistic drought quantification. Over the past few decades, numerous precipitation datasets have been developed based on gauge, reanalysis and satellite data. Nevertheless, differences in annual mean global precipitation between datasets can be up to 300 mm yr$^{-1}$ and the error can reach up to 100 mm per month when compared with gauge observations[29,30]. Finally, it is necessary to mention that drought assessments depend on the selected index and calculation methodology. For example, selecting a calibration period for drought index models such as the Palmer Drought Severity Index can significantly influence global drought trend interpretation, amplifying extreme drought areas by up to 15% (ref. 28). Overall, uncertainties in datasets, methods and model structure introduce substantial uncertainty in assessing drought and its trends, as highlighted in the Sixth Assessment Report of the Intergovernmental Panel on Climate Change[13,20].

Here, given the existing critical priority of reducing uncertainties in the quantification of recent trends in drought severity, we used the most accurate global precipitation datasets[29,30] and computed AED using the comprehensive Penman–Monteith method. For our drought index model, we applied the Standardized Precipitation Evapotranspiration Index (SPEI)[31], which balances complexity and utility by effectively representing the supply–demand dynamics of drought through the difference between precipitation and AED, allowing spatial and temporal comparability and quantification of the sensitivity of the index to variations of AED in different world regions and climate conditions[32]. Moreover, the SPEI method generates estimates of drought variability across multiple timescales (1–48 months) without requiring a calibration period, which allows an objective assessment of the recent trends in drought severity and quantification of the influence of increased AED. Numerous studies have analysed drought trends at the regional and national scales using SPEI, demonstrating its ability to identify drought trends linked to anthropogenic forcing[33,34]. Although some studies have explored drought projections using SPEI[35,36], only a few have examined global-scale trends, indicating an increase in drought severity associated with global warming[37]. Other global studies have assessed drought trends using SPEI with observational data but did not evaluate the influence of AED on drought severity or address uncertainties in precipitation and AED datasets—critical limitations for drawing robust conclusions[17,38,39]. Only one study[2] has examined the role of anthropogenic climate change on drought severity using Coupled Model Intercomparison Project Phase 6 simulations, but it introduces significant uncertainties owing to the limitations of model-based approaches. Although SPEI has been widely used to assess drought trends, this study quantifies, at the global scale and based on observations, the role of increasing AED in drought severity. In addition, it evaluates uncertainties in global datasets, offering a more comprehensive perspective on this critical issue.

## Global drought trends

We developed 4 global, high-resolution (0.05°) SPEI datasets for 1981–2022 using precipitation from Climate Hazards Group Infrared Precipitation with Station Data (CHIRPS)[40] or Multi-Source Weighted-Ensemble Precipitation (MSWEP)[41], combined with AED from the Global Land Evaporation Amsterdam Model version 4.2a (GLEAM)[42] or hourly potential evapotranspiration (hPET)[43]. Although both precipitation products perform well[29,30], the inputs and methods used to produce CHIRPS and MSWEP are quite different. Similarly, the widely used GLEAM and hPET AED datasets rely primarily on satellite and reanalysis data sources. Hence using combinations of all four builds a robust foundation for assessing trends. To assess global trends before the 1980s, we also developed two additional SPEI datasets based on ERA5-Land reanalysis (the fifth-generation reanalysis from the European Centre for Medium-Range Weather Forecasts, ERA5; about 25 km) and the Climatic Research Unit Time-Series (CRU-TS;

about 50 km), covering 1950–2022 and 1901–2022, respectively. By incorporating multiple datasets and different periods, we aim to capture a broader range of potential uncertainties in the forcing data and provide a more comprehensive assessment of drought patterns. Through using climatological AED and precipitation, we developed equivalent datasets that enable us to quantify the contributions of AED and precipitation changes to the SPEI trend, as well as to the frequency, duration and magnitude of drought events. Here we focus on the 6-month SPEI, as it captures prevalent short- to medium-term drought conditions.

On the basis of the mean of the four high-resolution SPEI datasets (HRSPEI) datasets, the quasi-global average (50° S to 50° N) 6-month SPEI shows a decreasing trend, indicating drying conditions during the period 1981–2022 (Fig. 1). The 6-month HRSPEI shows a significant ($P < 0.05$) decreasing trend of −0.0055 ± 0.002 yr$^{-1}$ (Fig. 1a). The quasi-global area in drought (SPEI < −1) shows a commensurate significant increasing trend of 0.36 ± 0.03% yr$^{-1}$ (Fig. 1b). For severe (SPEI < −1.4) and extreme (SPEI < −1.8) droughts, the area in drought shows a significant increasing trend of 0.17 ± 0.02% yr$^{-1}$ and 0.047 ± 0.022% yr$^{-1}$, respectively. On the basis of CRU-TS and ERA5, the period from 1950 to 1980 shows significant increasing trends in 6-month SPEI of 0.00120 z-units yr$^{-1}$ and 0.012 z-units yr$^{-1}$, respectively. A summary of the 6-month SPEI trend is provided in Extended Data Fig. 1f.

Spatially, the 6-month HRSPEI shows a drying trend across large parts of the world such as in Europe, Africa, western North America and South America during 1981–2022 (Fig. 1c), with a drying trend of up to −0.08 z-units yr$^{-1}$. Conversely, regions such as South and Southeast Asia, the Guyanas in South America, central Southern Africa and eastern North America show an increasing wetting trend over the same period. The trends for individual datasets that constitute the HRSPEI and CRU-TS and ERA5 datasets are provided in Extended Data Figs. 1 and 2, respectively.

The trend in magnitude and frequency of droughts has increased in different parts of the world during 1981–2022 (Fig. 2). The drought magnitude (Fig. 2a) and frequency (Fig. 2b) show significant decreasing and increasing trends in various regions, particularly in the southern parts of South America, eastern and central Africa, southern Europe and the western United States. Compared with much of the world, parts of Africa and South America show a greater increase and decrease in drought frequency and magnitude, respectively, highlighting that these trends are primarily driven by precipitation deficits. In contrast, changes in drought duration are statistically significant only in scattered areas, arguing against a widespread change in drought duration (Extended Data Fig. 3).

Of note is the acceleration in the decrease in SPEI and increase in areas experiencing drought during the past 5 years, with 2022 recording the highest percentage of impacted areas (Extended Data Fig. 4). During this period, the global extent of severe and extreme drought increased threefold and fivefold, respectively, compared with 1981–2022. In Europe, 82% of land experienced drought, with 50% under moderate to severe drought (Fig. 1d). In 2022, annual precipitation across Europe dropped by up to 35% below the 1981–2022 average, and AED increased by up to 40% (Extended Data Fig. 5).

## Drivers of changes in drought

To assess how changes in AED and precipitation affect drought, we compare SPEI trends calculated from observed AED and precipitation variations with those based on climatological means of AED (AEDclm) and precipitation (Prclm). The quasi-global average 6-month SPEI trend, based on observed precipitation and AEDclm, is 0.002 z-units yr$^{-1}$, which is about 131% higher than the observed trend (Fig. 3), indicating that holding AED to its climatological value results in a positive trend. When using observed AED and Prclm, the SPEI trend is −0.02 z-units yr$^{-1}$, which is 300% more negative than the observed trend (Fig. 3a). Similarly, the

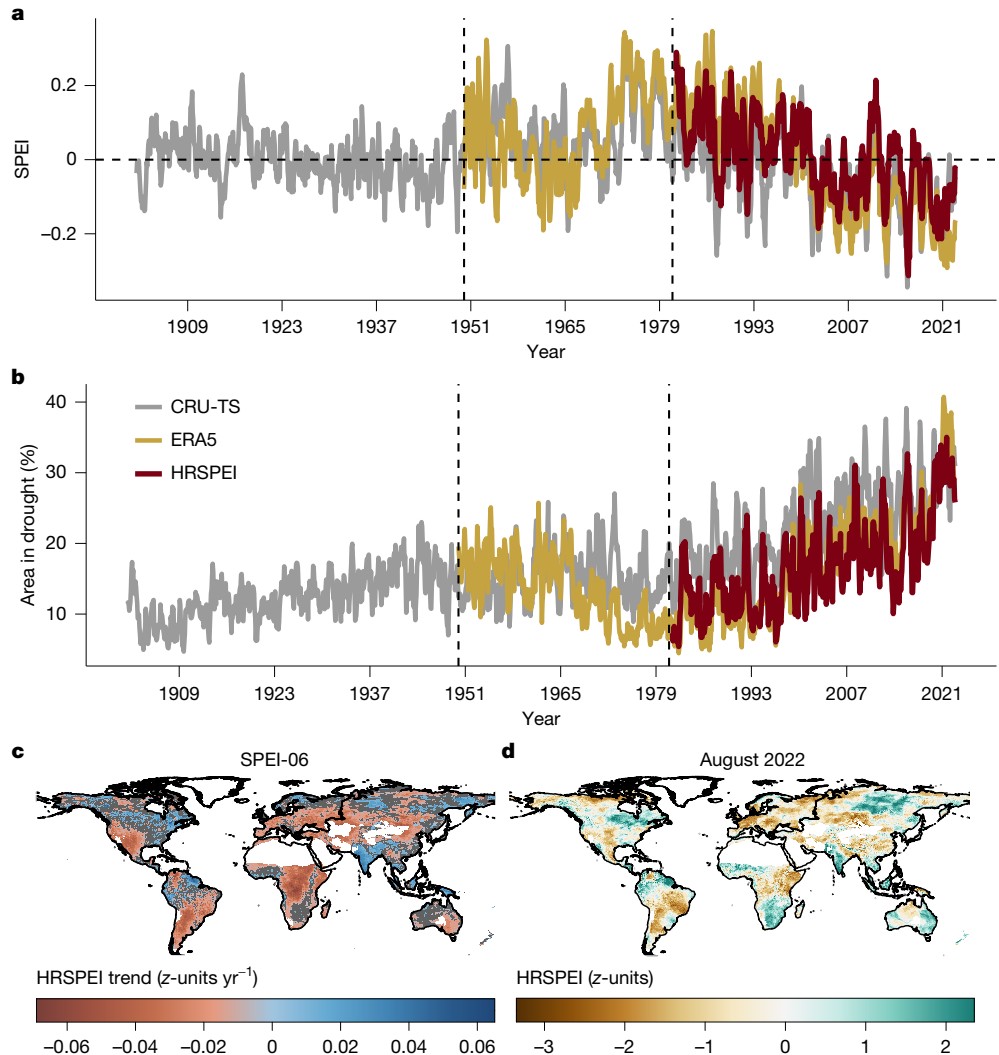

**Fig. 1 | Monthly SPEI, percentage of area in drought and maps of SPEI trends and the 2022 drought. a,b**, The quasi-global (50° S to 50° N) average HRSPEI (**a**) and global percentage of area in droughts (**b**). **c**, The trend in 6-month HRSPEI for 1981–2022 (*z*-units yr$^{-1}$), with non-significant trends ($P > 0.05$) marked in grey for visualization. **d**, The 6-month HRSPEI values for the record-breaking drought in August 2022 (*z*-units). The time series uses HRSPEI (0.05°), CRU-TS (0.5°) and ERA5 (0.25°), with HRSPEI being the ensemble mean of MSWEP_hPET,

MSWEP_GLEAM, CHIRPS_hPET and CHIRPS_GLEAM (1981–2022). CRU-TS covers 1901–2022 and ERA5 spans 1950–2022. The time series are averaged over tropical and subtropical land areas (50° S to 50° N), excluding regions with average annual rainfall below 180 mm. For regions above 50° N, the spatial trend is based on the mean of MSWEP_hPET and MSWEP_GLEAM, as CHIRPS is available up to 50° N. The vertical lines indicate the period from 1950 to 1980, showing higher positive SPEI values based on ERA5 and CRU-TS compared with 1981–2022.

trend in areas in drought based on the observed precipitation and AED-clm is −0.004% yr$^{-1}$, which is 96% lower than the observed trend. These findings indicate that AED changes from 1981 to 2022 intensified both the downwards trend in SPEI and the expansion of drought-affected areas. The time series based on Prclm shows an evolution from positive SPEI values at the beginning of the study period to negative in recent years (Fig. 3a). This pattern is also observed with the SPEI values based on ERA5 (Extended Data Fig. 6), highlighting the increased impact of AED as precipitation remains fixed at its climatological value.

Regionally, the results indicate a notable contribution of AED to the negative SPEI trend (up to −0.06 yr$^{-1}$) in large parts of Europe (excluding Norway and Sweden), Asia, Australia, the western United States and southern parts of South America (Fig. 3b). In addition, in parts of East and South Africa, changes in AED have exacerbated the negative SPEI trend by up to −0.04 *z*-units yr$^{-1}$. In contrast, AED has minimal or no effect on drought trends in North America (Canada, Midwest and Southeast United States), northern South America (Amazon River Basin) and Central Africa. However, AED appears to have increased the SPEI trend (up to +0.02 *z*-units yr$^{-1}$) in South (India) and

Southeast Asia. This change can be attributed to the observed increasing trend in precipitation and decreasing trend in AED (Extended Data Fig. 7). When using Prclm, the 6-month SPEI shows a significantly more negative trend (up to −0.1 *z*-units yr$^{-1}$) compared with the observed trend globally, except in South and Southeast Asia (Fig. 3c). The trend based on ERA5 datasets also shows a similar change during 1981–2022 (Extended Data Fig. 6).

Observed changes in AED have also intensified the magnitude and frequency of droughts globally (Fig. 2). Compared with AEDclm, observed trends show a more negative drought magnitude (up to −0.2 *z*-units yr$^{-1}$) and a more positive frequency trend (up to +0.16 months yr$^{-1}$). Regional averages reveal that drought magnitude, based on observed AED, shows a significant decreasing trend between −0.1 yr$^{-1}$ and −0.05 yr$^{-1}$, whereas the trend is not statistically significant with AEDclm in South and North America, Africa, Europe, and Australia (Fig. 2g–r). Drought frequency shows a significant increasing trend between 0.02 months yr$^{-1}$ and 0.07 months yr$^{-1}$ with observed AED, whereas the trend is very low and not significant using AEDclm. In Asia, AEDclm shows a significant increase in drought magnitude (0.03 *z*-units yr$^{-1}$) and a decrease in

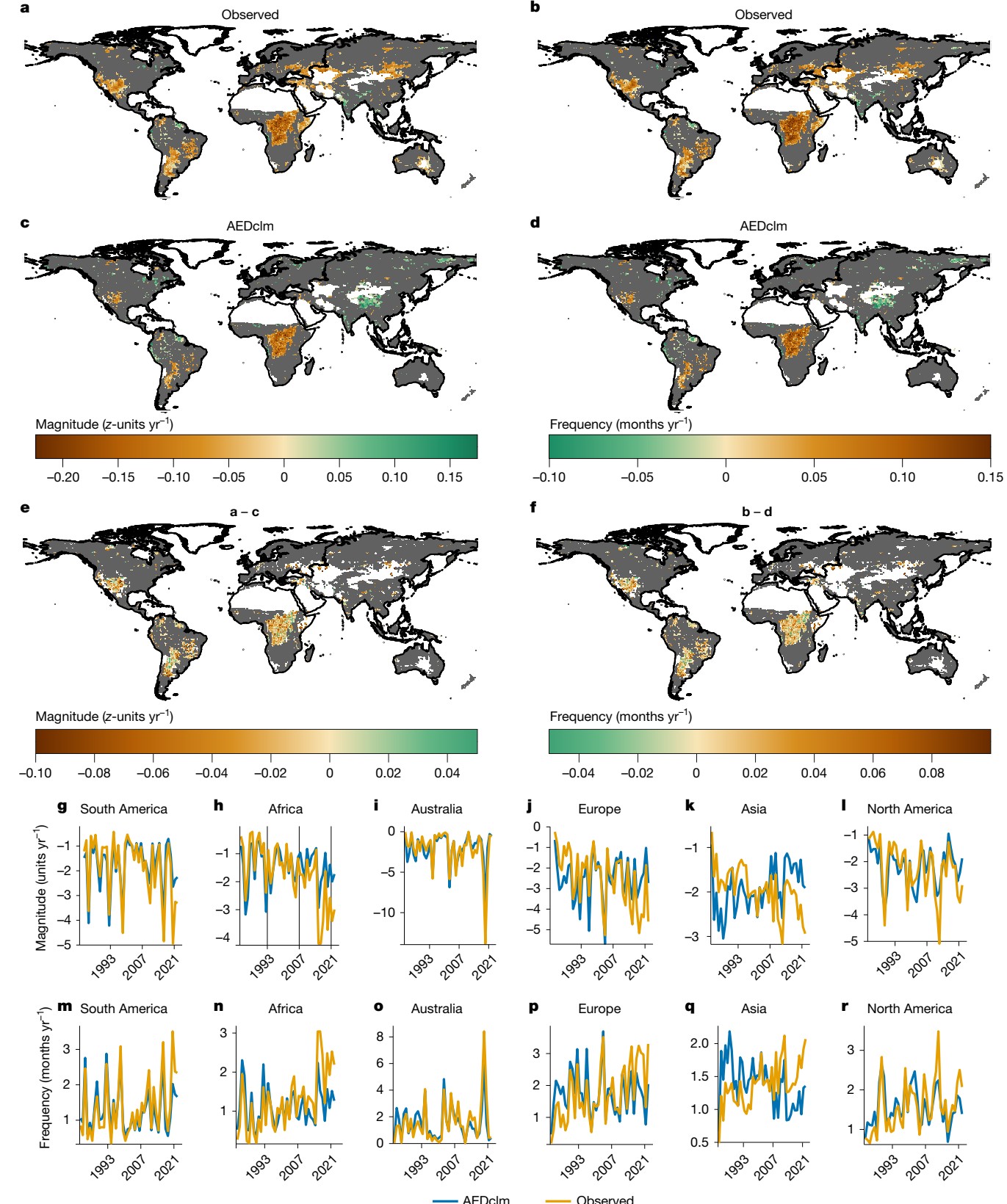

**Fig. 2 | Trends in drought magnitude and frequency for 6-month SPEI during 1981–2022. a,b**, The trend in magnitude (*z*-units yr⁻¹; **a**) and frequency (months yr⁻¹; **b**) of droughts (SPEI < −1) for the period 1981–2022 based on observed precipitation and AED ('Observed'). **c,d**, The trend in magnitude (**c**) and frequency (**d**) based on observed precipitation and AEDclm ('AEDclm'). **e,f**, The difference in trend between observed precipitation and AEDclm for drought magnitude (**e**) and frequency (**f**). The SPEI is based on MSWEP_hPET.

The trend and regional average exclude dry land areas with average annual rainfall below 180 mm. Non-significant trends (*P* > 0.05) are marked in grey to enhance clarity. Magnitude is calculated as the cumulative sum of SPEI < −1 values during a drought event for each year and frequency represents the number of events in a year with SPEI < −1. **g–r**, The average magnitude (units yr⁻¹; **g–i**) and frequency (months yr⁻¹; **m–r**) of droughts averaged over South America (**g,m**), Africa (**h,n**), Australia (**i,o**), Europe (**j,p**), Asia (**k,q**) and North America (**l,r**).

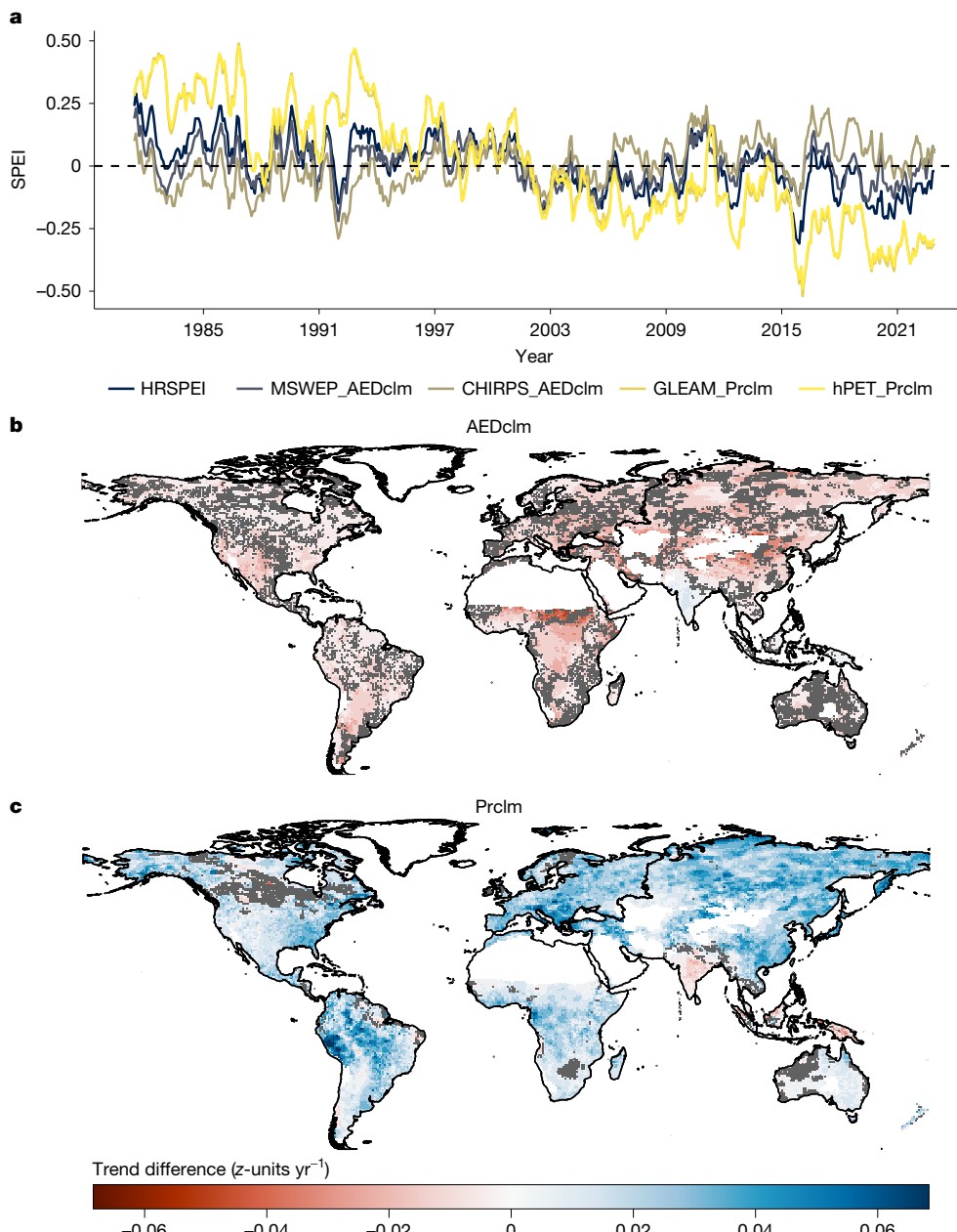

**Fig. 3 | Monthly time series and trend differences of 6-month SPEI during 1981–2022. a**, The quasi-global average (50° S–50° N) 6-month SPEI based on AEDclm, Prclm and HRSPEI. MSWEP_AEDclm and CHIRPS_AEDclm refer to the average SPEI based on MSWEP and CHIRPS precipitation and AEDclm (climatological mean of GLEAM and hPET). GLEAM_Prclm and hPET_Prclm show the average SPEI based on AED from GLEAM and hPET and Prclm (climatological mean of MSWEP and CHIRPS). **b**, The trend difference between SPEI based on observations (observed precipitation and AED) and SPEI based on observed precipitation and climatology of AED (AEDclm). **c**, The trend difference between SPEI based on observations and SPEI based on observed AED and climatology of precipitation (Prclm). Non-significant trends ($P > 0.05$) are marked in grey to enhance clarity. The trend excludes dry land areas with average annual rainfall below 180 mm. For regions above 50° N, the trend is based on the mean of MSWEP_hPET and MSWEP_GLEAM, as CHIRPS is available up to 50° N.

frequency (−0.02 months yr$^{-1}$). In contrast, observed AED indicates a decrease in magnitude (−0.03 $z$-units yr$^{-1}$) and an increase in frequency (0.02 months yr$^{-1}$).

Overall, even though precipitation accounts for 60% of the global average SPEI trend during 1981–2022, the role of AED, contributing 40%, is substantial (Fig. 4). This is especially notable considering the stronger sensitivity of SPEI to precipitation than to AED in most land regions[32]. In Africa, Australia, and the drylands of North and South America, the influence of AED is particularly pronounced, contributing up to 65% to drought trends during 1981–2022. Specifically, AED accounts for 44% of the drought trend in Africa and 51% in Australia, playing a significant role in intensifying drought severity in these regions. In contrast, the contribution of AED to drought trends in North and South America, Europe, and Asia is around 30%.

## Acceleration of droughts

The area affected by drought has expanded significantly, particularly during the past 5 years (Extended Data Fig. 4). Globally, during the past 5 years (2018–2022), the observed area in drought was on average 27%, which is 74% higher than during 1981–2017 and 58% higher compared with AEDclm for 2018–2022. Regionally, drought-affected areas increased by 119% in Australia, 163% in southern South America, and 141% in the western United States in 2018–2022 compared with

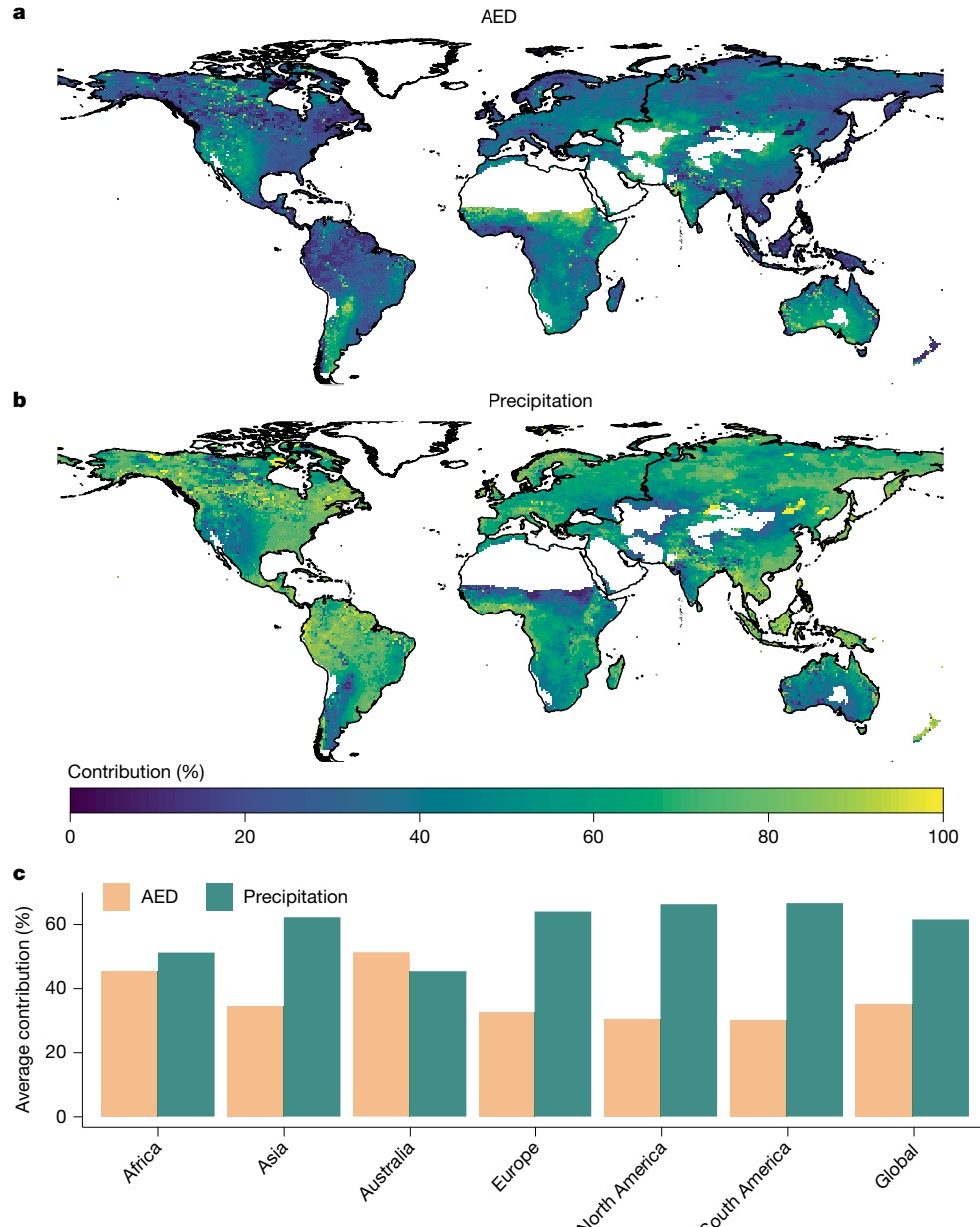

**Fig. 4 | Percentage contribution of AED and precipitation to trends in 6-month SPEI. a,b**, The percentage contribution of AED (**a**) and the percentage contribution of precipitation (**b**) to the observed changes in 6-month HRSPEI during 1981–2022. The contributions are computed by calculating the difference between the observed trend and the trend based on the climatological values of AED (AEDclm) and precipitation (Prclm). The contribution of AED is determined by the difference between the trend using observed AED and precipitation and the trend using observed precipitation and AEDclm. Similarly, the contribution of precipitation is calculated as the difference between the trend using observed precipitation and AED and the trend using observed AED and Prclm. The percentage contribution of each factor is then calculated as the absolute value of the difference divided by the total absolute difference, providing a relative measure of each factor's influence on the observed trend. **c**, The regional and global average contribution of precipitation and AED to the changes in SPEI.

1981–2017 (Extended Data Fig. 8). Similarly, in the past 5 years, drought areas increased by 75%, 80% and 56% in East Africa, Northern Asia and Europe, respectively. In contrast, when using AEDclm, the increases were substantially lower in Australia (36%), southern South America (62%), western United States (58%) and Northern Asia (0.5%), whereas Europe and East Africa experienced a decrease of about 8%. A summary of these changes is provided in Extended Data Fig. 8h.

Drought severity in 2022 was record-breaking relative to the 1981–2022 period (Extended Data Fig. 8). The year 2022 had the highest drought area (30%), which is 42% higher than AEDclm. As shown in Fig. 1d, the 6-month SPEI for August 2022 indicates moderate to extreme droughts across Europe, East Africa, western United States and southern South America, with drought-affected areas approximately 34–67%

greater than AEDclm. In addition, the average SPEI was −0.85 units yr$^{-1}$, compared with 0.52 units yr$^{-1}$ based on AEDclm. Overall, owing to the observed increase in AED, the trends in SPEI and areas in drought during 1981–2022 indicate that not only are drier regions becoming drier but also wet areas are experiencing drying trends.

## Discussion

According to the SPEI, over the past 42 years (1981–2022), global drought severity has intensified. In the past 5–10 years, this trend has accelerated as a consequence of the strong increase in AED, which is directly related to global warming and an increased vapour pressure deficit[18], as the water supply to the atmosphere is not enough to

compensate for the large temperature increase. Some recent studies have also suggested an increase in the severity of drought events over large land areas based on metrics such as modelled soil moisture[11] and the Palmer Drought Severity Index[44,45], all of which are sensitive to changes in the AED. Nevertheless, in our study, we have quantified the contribution of AED to worsening drought conditions, which has been up to 60% in some regions, particularly in Africa, Australia, western United States and southern South America. Moreover, changes in AED have exacerbated the drying trend globally, particularly in the past decade. The year 2022 specifically was a record-breaking year for drought severity and extent in Europe and East Africa. In Europe, the severity of the 2022 drought event can be largely attributed to anthropogenic warming, as the anomalies observed in streamflow and soil moisture cannot be explained by the precipitation deficit alone, but mostly by enhanced AED, which increased water losses by evaporation[24,25]. Moreover, the ecological drought severity recorded in Europe's natural forests cannot be fully explained without considering the influence of high temperatures and AED on plant physiology. In the absence of formal attribution studies in other regions of the world that experienced drought in 2022, the attribution in Europe and the increase in severity globally driven by enhanced AED as shown in this study suggests that it is reasonable to conclude that anthropogenic global warming likely contributed to exacerbate global drought severity in 2022.

Compared with previous studies analysing recent drought trends based on atmospheric drought indices that use AED in calculations[2,17,44,45], this study has isolated the effect of AED on drought severity and in addition our study has also reduced uncertainties given the use of high-spatial-resolution and multi-source data, which allows for a clearer understanding of drought intensification. The observed increase in drought severity aligns with associated impacts on agricultural, environmental and hydrological systems, as seen in events like the 2022 European drought, which contributed to enhanced tree mortality, increased forest fires and long-term soil moisture decline[11,46]. Although the SPEI is an atmospheric drought index that effectively captures the effects of precipitation and AED on drought severity, it may represent drought-related impacts very effectively[47]. However, further studies are needed, considering variables such as soil moisture, vegetation stress and hydrological flows for better understanding of the broader impacts of the observed changes on ecosystems and human activities[17]. Moreover, the observed acceleration of drought trends in the past few years aligns with future climate projections that indicate further increases in drought severity owing to projected warming[35,48], which warns of the need for better socioeconomic and environmental adaptation measures to reduce drought impacts and improve global drought.

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

## Methods

### Drought index

The SPEI[31] is a widely utilized drought assessment tool that incorporates both AED and precipitation to evaluate drought severity across different timescales. SPEI values are computed by subtracting AED from precipitation. These differences are standardized using a log-logistic probability distribution to ensure consistency across regions, seasons and timescales. This distribution model involves three parameters ($\alpha$, $\beta$ and $\gamma$), which are estimated using the L-moment procedure. The SPEI indices were calculated using the entire 1981–2022 period as a baseline, ensuring that the full range of variability in the input data is captured. Unlike other drought indices, the SPEI does not require a predefined baseline or calibration period, as it standardizes the data directly from the input time series, ensuring consistency across datasets and timescales. The SPEI values provide categories for wet and dry events (Extended Data Table 1).

Using SPEI, we developed four high-resolution SPEI indices using a combination of two precipitation datasets and two potential evapotranspiration (that is, AED) datasets. The precipitation datasets used were the MSWEP[41] and CHIRPS[40] precipitation and the AED datasets were GLEAM[42] and hPET[43]. The resulting four indices: MSWEP_GLEAM, MSWEP_hPET, CHIRPS_GLEAM and CHIRPS_hPET, were developed at a spatial resolution of 0.05° for the period 1981–2022. The 0.1°-resolution datasets were first interpolated to match the resolution of CHIRPS using bilinear interpolation. In addition, we developed an ensemble mean (HRSPEI) based on all four datasets. For latitudes above 50° N, the mean is derived from MSWEP_GLEAM and MSWEP_hPET, as CHIRPS data are available only up to ±50° latitude. AED and AED variability in high-latitude areas >50° N are generally small, and changes in AED, even at high percentages, result in low absolute magnitudes, making SPEI less sensitive to AED in these regions[32].

To assess the contributions of precipitation and AED, we developed additional indices based on observed (that is, actual values from hPET and GLEAM) AED with monthly climatological precipitation (Prclm), and observed (that is, a combination of gauge and satellite and reanalysis data) precipitation with climatological AED (AEDclm) for the period 1981–2022. Using AEDclm and Prclm allows us to quantify the impact of precipitation and AED changes and variability on droughts over the past 42 years. To further assess changes in drought during the early and mid-1990s, we developed two coarse-resolution SPEI indices based on ERA5 (0.25°) and CRU-TS (0.5°). The SPEI based on ERA5 was computed using monthly precipitation and AED derived from ERA5 meteorological datasets using the Penman–Monteith equation (equation (1)) for the period 1950–2022. Similarly, the SPEI based on CRU-TS was calculated using monthly precipitation and AED derived from CRU-TS meteorological datasets using the Penman–Monteith equation (equation (1)) for the period 1901–2022.

In this study, we use SPEI < −1 as the threshold to define a drought, with values between −1 and 1 considered near-normal conditions and values >1 indicating wet conditions (Extended Data Table 1). Using SPEI < −1 values, we assessed key drought metrics: magnitude, duration, intensity and frequency. We follow the classic approach and widely adopted methods to define these metrics[49]. Drought magnitude is calculated as the cumulative sum (running total) of SPEI < −1 values during a drought event. Drought intensity is defined as the maximum negative value of SPEI observed during the event. Duration represents the run length of consecutive months with SPEI < −1, and frequency is the total number of drought events within a given period[49]. Finally, severity is used as an overarching term to refer to all aspects of drought: intensity, magnitude, duration and extent.

### Global climate and AED datasets

The MSWEP (version 2.8) dataset offers global 3-hourly, daily and monthly precipitation estimates at a 0.1° spatial resolution from 1979 to present[41]. Similarly, the CHIRPS (version 2.0) dataset provides daily, decadal and monthly precipitation estimates over land, with a spatial resolution of 0.05° for latitudes below 50°, covering the period from 1981 to present[40]. Both MSWEP and CHIRPS are high-resolution precipitation datasets developed by integrating ground-station observations, satellite data and reanalysis products.

CHIRPS and MSWEP were chosen as they generally outperform other similar gridded precipitation datasets when compared with ground observations[29,30]. CHIRPS (0.05°) is particularly designed for monitoring droughts and detecting environmental changes, providing daily precipitation estimates from 1981 to present. It combines satellite-derived Climate Hazards Center Infrared Precipitation (CHIRP) and the Climate Hazards Group Precipitation Climatology (CHPclim) with ground-station data from the Global Historical Climate Network and many other sources. The CHIRPS product benefits from a high degree of homogeneity, provided by its simple but consistent foundation of geostationary thermal infrared satellite observations. CHIRPS also incorporates unique observation inputs from Africa, Latin America and Central America. MSWEP (0.1°) has been designed with both accuracy and homogeneity in mind, providing 3-hourly precipitation estimates from 1979 to present. It integrates daily observations from over 77,000 stations from various national and international data sources, satellite estimates from infrared- and microwave-based satellite datasets, and reanalysis data, offering accurate global precipitation data from 1979 to present. Both CHIRPS and MSWEP have previously been evaluated globally using statistical metrics such as Kling–Gupta efficiency and Nash–Sutcliffe efficiency, as well as various bias and error metrics[29,30]. For instance, MSWEP outperformed 22 other global precipitation datasets in capturing daily precipitation from 76,086 gauging stations and in driving hydrological models across 9,053 catchments[29]. In addition, both MSWEP and CHIRPS were found to outperform other high-resolution gauge-based datasets in modelling daily, monthly and annual streamflow across 1,825 streamflow gauges[30]. However, both datasets remain subject to inherent uncertainties, and, therefore, considering both helps reduce biases and obtain more reliable estimates, given that they are somewhat independent. For example, they differ in their data sources, with CHIRPS using only geostationary thermal infrared observations, whereas MSWEP also uses microwave observations, and they use different sets of station data to correct locally. Despite these differences, the monthly correlation between MSWEP and CHIRPS shows a high correlation across most regions, except for Central Asia (Extended Data Fig. 9a). The average monthly difference between the 2 datasets varies spatially, reaching up to ±40 mm (Extended Data Fig. 9d). Notably, larger discrepancies occur in regions such as the Amazon, Central Africa and parts of Southeast Asia. Such convergence between the two products helps reduce concerns about the uncertainties owing to different approaches and changes in the constellation of Earth-observing satellites that can affect the robustness of their representation of changes over time.

The hPET is a global hourly AED dataset developed using ERA5 climate datasets and the Food and Agriculture Organization (FAO)'s Penman–Monteith equation (equation (1)). hPET is available for the global land surface at 0.1° spatial resolution covering the period 1981–2022[43]. In addition, the AED from GLEAM (version 4.2a) is a global dataset derived using Penman's original equation (equation (2)), using satellite and reanalysis datasets[42]. GLEAM is available at a 0.1° spatial resolution and covers the period 1980–2023. hPET is based on the FAO Penman–Monteith equation, which computes reference crop evaporation by assuming certain surface and aerodynamic characteristics that are constant in time. In contrast, GLEAM calculates aerodynamic conductance as a dynamic variable depending on ecosystem characteristics and local meteorology and therefore is space and time dependent. Nonetheless, given the dominant influence of radiative forcing and atmospheric aridity in both computations, their estimates are overall similar. The correlation between GLEAM and hPET exceeds 0.9 across 91% of the

global land surface (Extended Data Fig. 9b), and the monthly average difference between them is up to ±3 mm (Extended Data Fig. 9c).

The global AED and precipitation data from the CRU-TS dataset are available at a spatial resolution of 0.5°, covering the period from 1901 to present[50]. Similarly, the ERA5 reanalysis dataset, representing the fifth-generation reanalysis from the European Centre for Medium-Range Weather Forecasts (ECMWF), is available at a spatial resolution of 0.25° from 1940 to present[51].

### Atmospheric evaporative demand

The hPET is estimated using the FAO-56 Penman–Monteith equation (equation (1)), and the GLEAM PET (potential evapotranspiration, AED) is calculated using Penman's equation, including aerodynamic conductance (equation (2)). In addition, the FAO-56 Penman–Monteith method is applied to calculate AED from ERA5 climate datasets for the period 1950–2022 and CRU-TS climate datasets for 1901–2022. The Penman and FAO-56 Penman–Monteith methods consider various meteorological variables such as wind speed, air temperature, radiation and humidity to estimate AED:

$$PET_{pm} = \frac{0.408\Delta \times (R_n - G) + \gamma \times \frac{900}{T+273} \times u_2(e_s - e_a)}{\Delta + \gamma(1 + 0.34 u_2)} \quad (1)$$

$$PET_p = \frac{\Delta(R_n - G) + \rho_a \times c_p \times g_a \times (e_s - e_a)}{\lambda_v \times (\Delta + \gamma)} \quad (2)$$

where $\Delta$ is the slope of the plot of saturation vapour pressure–temperature relationship, $R_n$ is the net radiation, $G$ is the soil heat flux, $\gamma$ is the psychrometric constant, $T$ is the mean daily air temperature at 2-m height, $u_2$ is the wind speed at 2-m height, $(e_s - e_a)$ is the vapour pressure deficit of the air (difference between saturation vapour pressure and actual vapour pressure), $\rho_a$ is the air density, $c_p$ is the specific heat capacity of air at constant pressure, $g_a$ is the aerodynamic conductance, and $\lambda_v$ is the latent heat of vaporization.

### Trend analysis

The trend in SPEI is assessed using the non-parametric Mann–Kendall test and Sen's slope estimator. The Mann–Kendall test identifies upwards or downwards trends in the SPEI time series for each pixel. Sen's slope estimator calculates the slope of change in the SPEI series by computing the median of all possible slopes between data points. This method provides a robust estimate of the trend, particularly in the presence of outliers or nonlinear patterns. To identify drought events at the pixel scale, we utilize SPEI categories (Extended Data Table 1). SPEI values less than −1.0 are used to identify areas affected by droughts. We evaluate the frequency, duration and magnitude of these drought events (SPEI < −1) by analysing the number of occurrences, the length of consecutive periods and the intensity of SPEI values during the period from 1981 to 2022.

### Data availability

The high-resolution SPEI datasets[52], developed using the Standardized Precipitation Evapotranspiration Index (SPEI)[31], are freely accessible through the Centre for Environmental Data Analysis (CEDA) at https://doi.org/10.5285/ac43da11867243a1bb414e1637802dec and on JASMIN at /badc/hydro-jules/data/Global_drought_indices. The CHIRPS data can be accessed via the Climate Hazards Group (CHG) at https://www.chc.ucsb.edu/data/chirps/ (ref. 40). The MSWEP precipitation dataset is available from the GloH2O website at https://www.gloh2o.org/mswep/ (ref. 41). The hPET dataset is hosted by the University of Bristol at https://data.bris.ac.uk/data/dataset/qb8ujazzda0s2aykkv0oq0ctp (ref. 43). The AED data from GLEAM can be accessed at https://www.gleam.eu/ (ref. 42). The CRU-TS precipitation and AED datasets are available through CEDA at https://data.ceda.ac.uk/badc/cru/data/cru_ts/cru_ts_4.08/ (ref. 50). The ERA5 dataset is available for download from the Copernicus Climate Change Service's Climate Data Store at https://cds.climate.copernicus.eu/datasets (ref. 51).

### Code availability

This study utilized the SPEI code to calculate drought indices. The SPEI code is publicly available on GitHub at https://github.com/sbegueria/SPEI/. For trend analysis, the Trend package in R was used, which is publicly available at https://github.com/cran/trend. The code used to develop the global SPEI datasets, perform the trend tests and produce the figures is available on Zenodo at https://doi.org/10.5281/zenodo.15073433 (ref. 53).

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

**Acknowledgements** We acknowledge the financial support provided by the UK Foreign, Commonwealth and Development Office (FCDO, grant number 201880) and the UK Natural Environment Research Council (NERC, grant number NE/S017380/1). We thank the Centre for Environmental Data Analysis (CEDA, https://www.ceda.ac.uk/) for hosting the global drought datasets.

**Author contributions** S.H.G. developed the drought datasets, conducted trend assessments and drafted the paper. S.J.D. led the project, and S.H.G. and S.J.D. designed the study with input from all authors. C.F., H.E.B., D.G.M. and M.B.S. provided and developed the CHIRPS, MSWEP, GLEAM and hPET datasets, respectively. S.M.V.-S. provided the SPEI index. J.S., S.M.V.-S., D.G.M. and J.P. contributed to the experimental design, research questions and dataset selection. E.D. and J.T. supported the data analysis and edited the paper. All authors contributed to the development of the paper.

**Competing interests** The authors declare no competing interests.

**Additional information**
**Correspondence and requests for materials** should be addressed to Solomon H. Gebrechorkos.

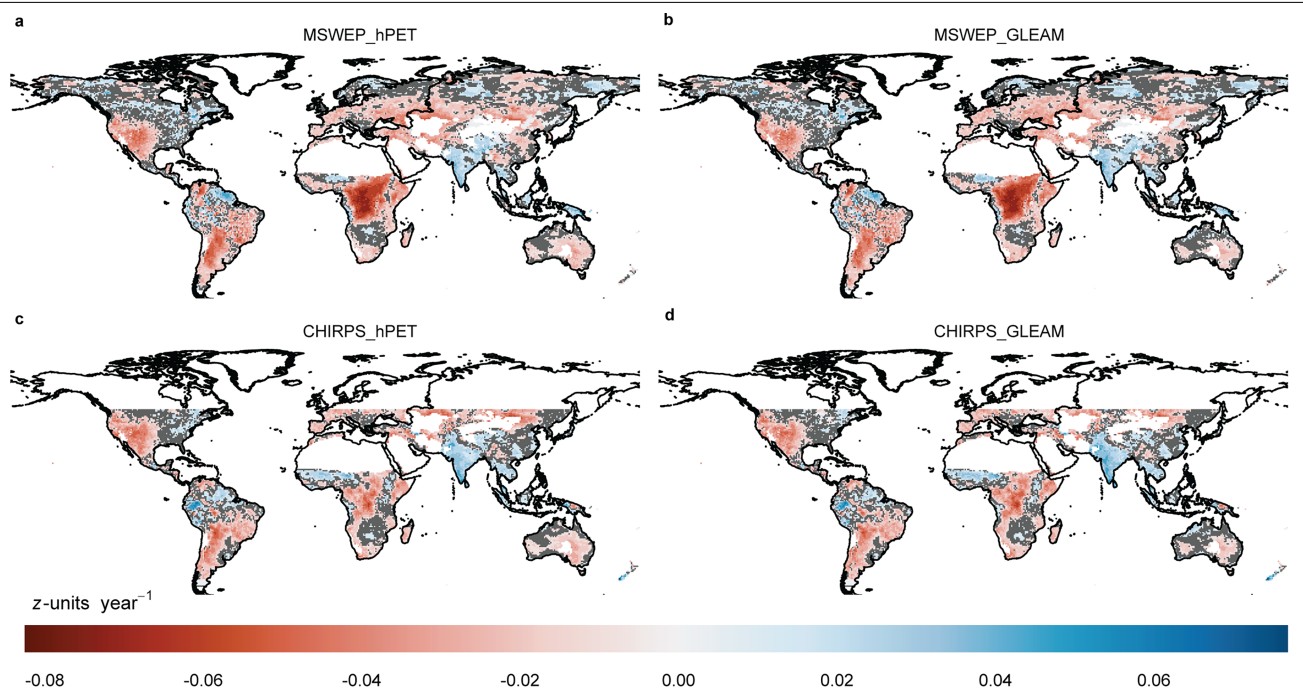

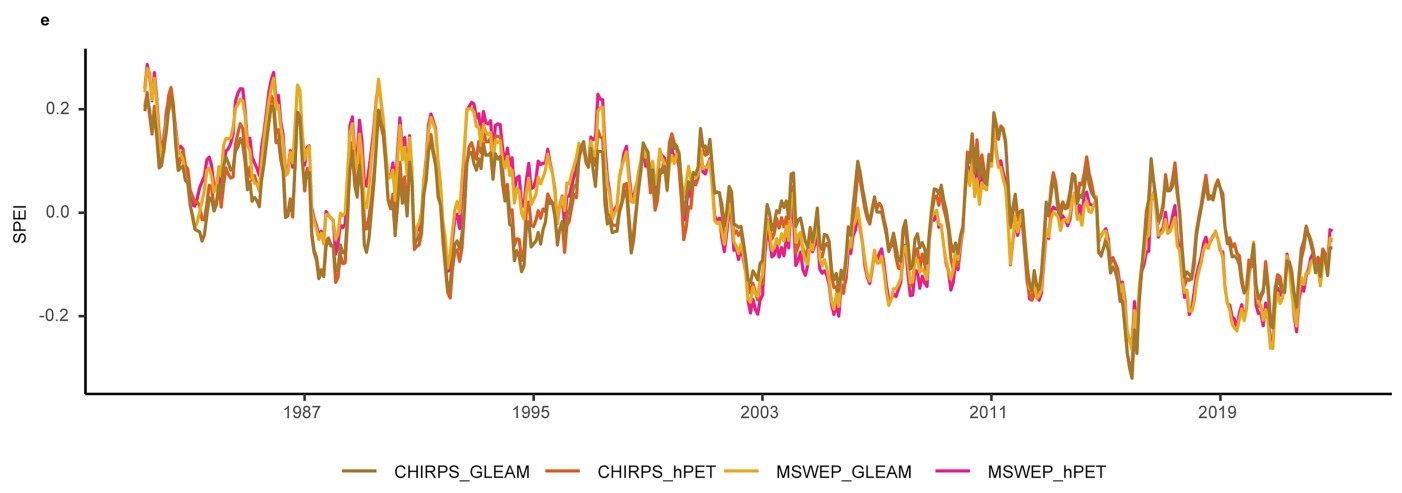

| Dataset | Period | Trend in SPEI (year⁻¹) | Trend in Area in Drought (% year⁻¹) |
|---------|--------|------------------------|--------------------------------------|
| HRSPEI | 1981-2022 | −0.0055 ± 0.002 | 0.36 ± 0.03% |
| CRU-TS | 1981–2022 | -0.004 | 0.33 |
| ERA5 | 1981–2022 | -0.01 | 0.52 |
| CRU-TS | 1950–1980 | 0.0013 | -0.04 |
| ERA5 | 1950–1980 | 0.0064 | -0.37 |

**Extended Data Fig. 1** | See next page for caption.

**Extended Data Fig. 1 | Trends in 6-month SPEI for the period 1981–2022.** Panels a), b), c), and d) display the 6-month SPEI trends (z-units year$^{-1}$) derived from the MSWEP_hPET, MSWEP_GLEAM, CHIRPS_hPET, and CHIRPS_GLEAM datasets, respectively. Non-significant trends (P-value > 0.05) are marked in gray to improve clarity. The analysis excludes dryland regions with an average annual rainfall of less than 180 mm. Panel e) shows the quasi-global (50°S to 50°N) average 6-month SPEI time series. Panel f) summarizes the SPEI trends and areas in drought based on HRSPEI, CRU-TS, and ERA5. The trend derived from HRSPEI represents the overall trend, while the deviation (±) reflects the spread in trends of the individual datasets (MSWEP_hPET, MSWEP_GLEAM, CHIRPS_hPET, and CHIRPS_GLEAM) around the HRSPEI mean trend. The deviation is calculated as the standard deviation of trends across these four datasets, highlighting the variability in the trends relative to the HRSPEI trend.

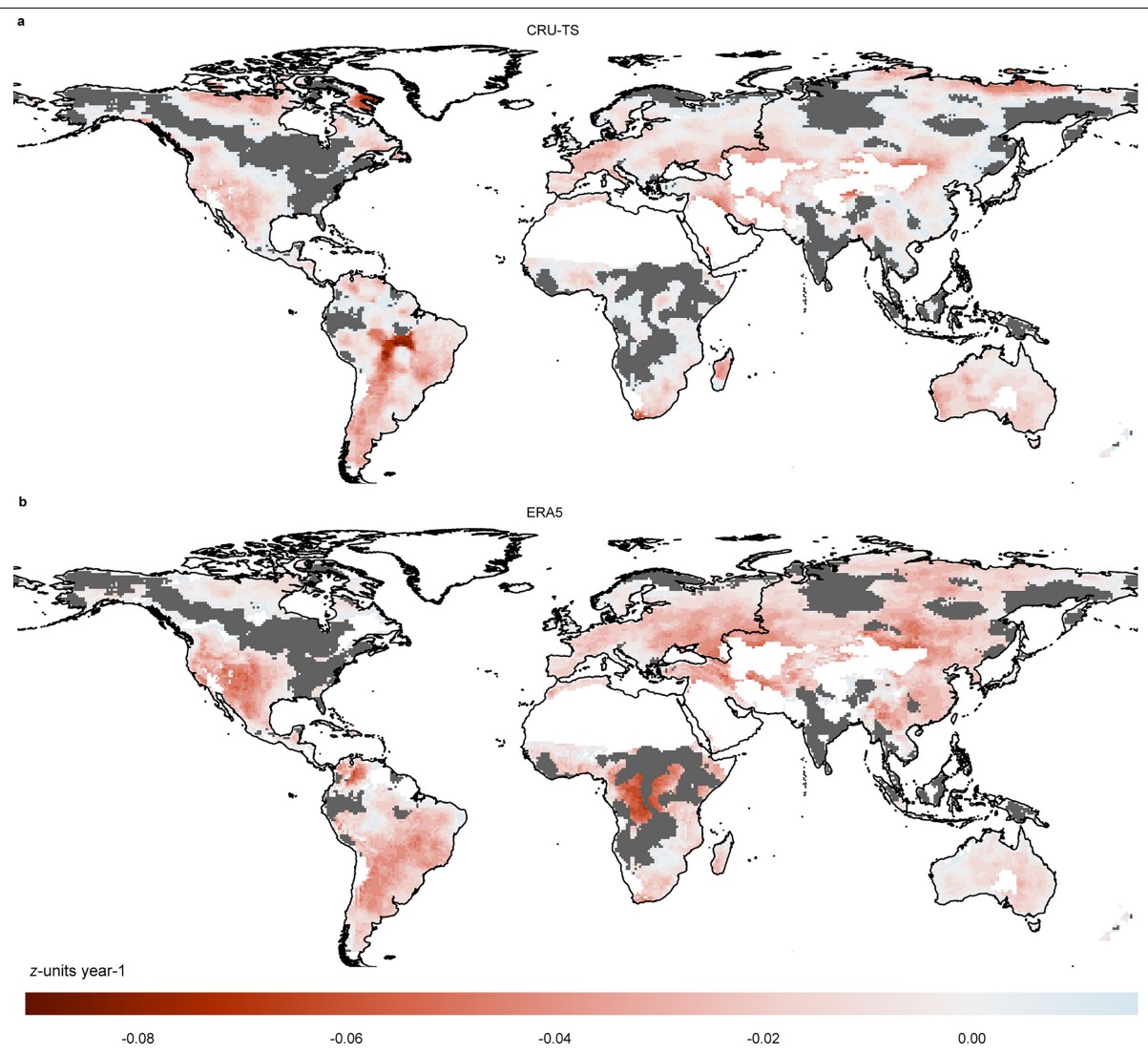

**Extended Data Fig. 2 | Trend in 6-month SPEI based on CRU-TS and ERA5 during 1981–2022.** Panel a) shows the trends in 6-month SPEI using precipitation and AED from the CRU-TS dataset. Panel b) illustrates the trends in 6-month SPEI derived from precipitation and AED based on ERA5 datasets. Non-significant trends (P-value > 0.05) are marked in gray to improve clarity. The analysis excludes dryland regions with an average annual rainfall of less than 180 mm.

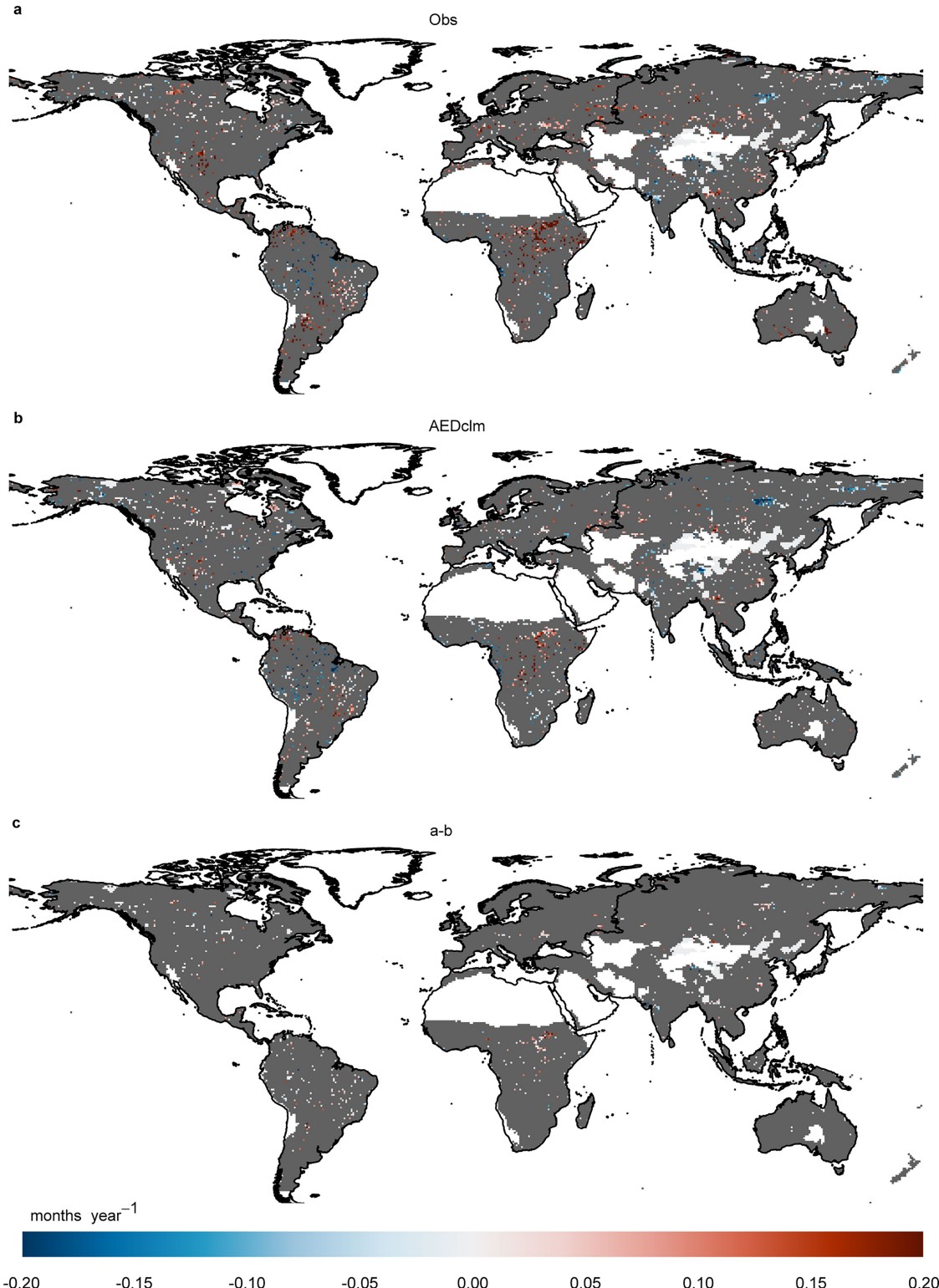

**Extended Data Fig. 3 | Trends in the duration of 6-month SPEI droughts.** The trends in 6-month SPEI are based on a) observed AED and precipitation (Obs) and b) climatological AED and observed precipitation (AEDclm). Panel c) shows the difference between the drought trends based on Obs and AEDclm. The SPEI is based on MSWEP_hPET, with drought duration calculated for each year where SPEI values fall below −1. Non-significant trends (P-value > 0.05) are marked in gray for clarity and the trend excludes dry land areas with an average annual rainfall below 180 mm.

a

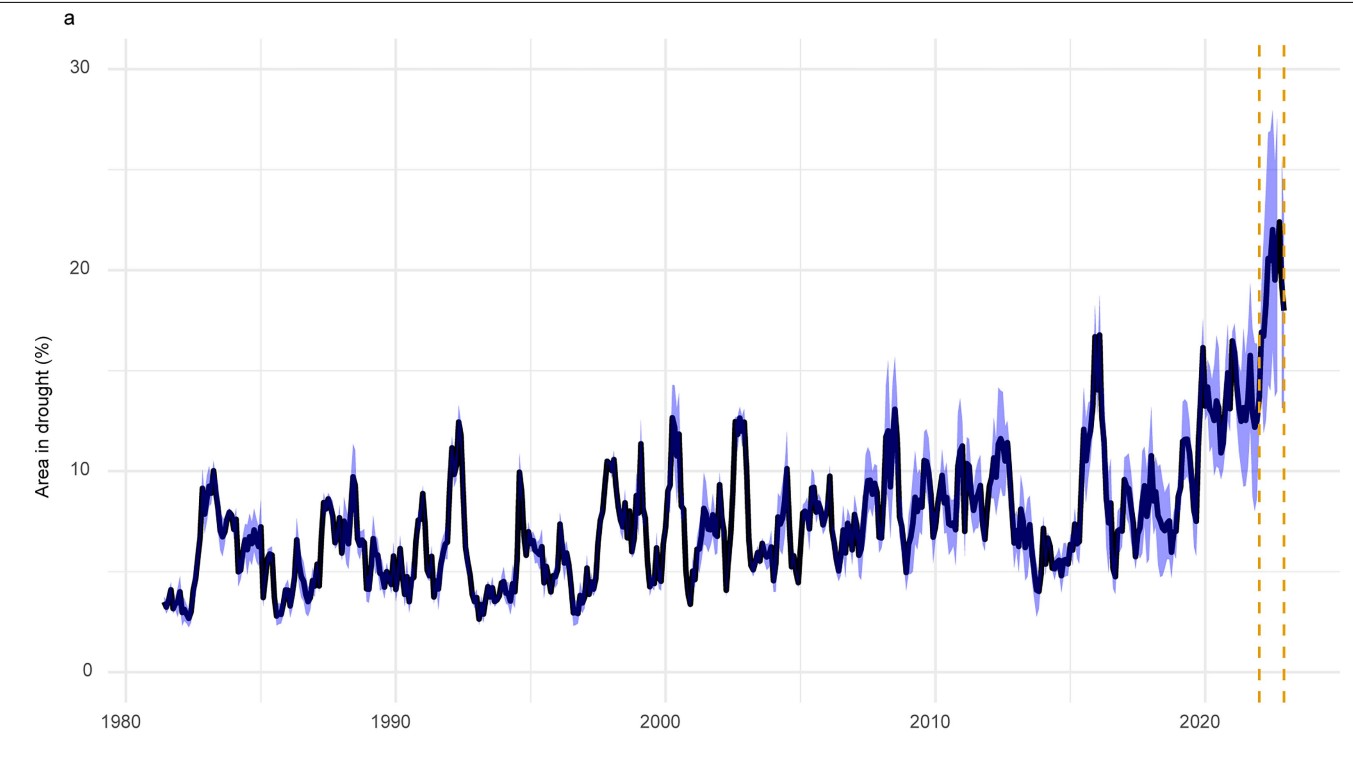

b

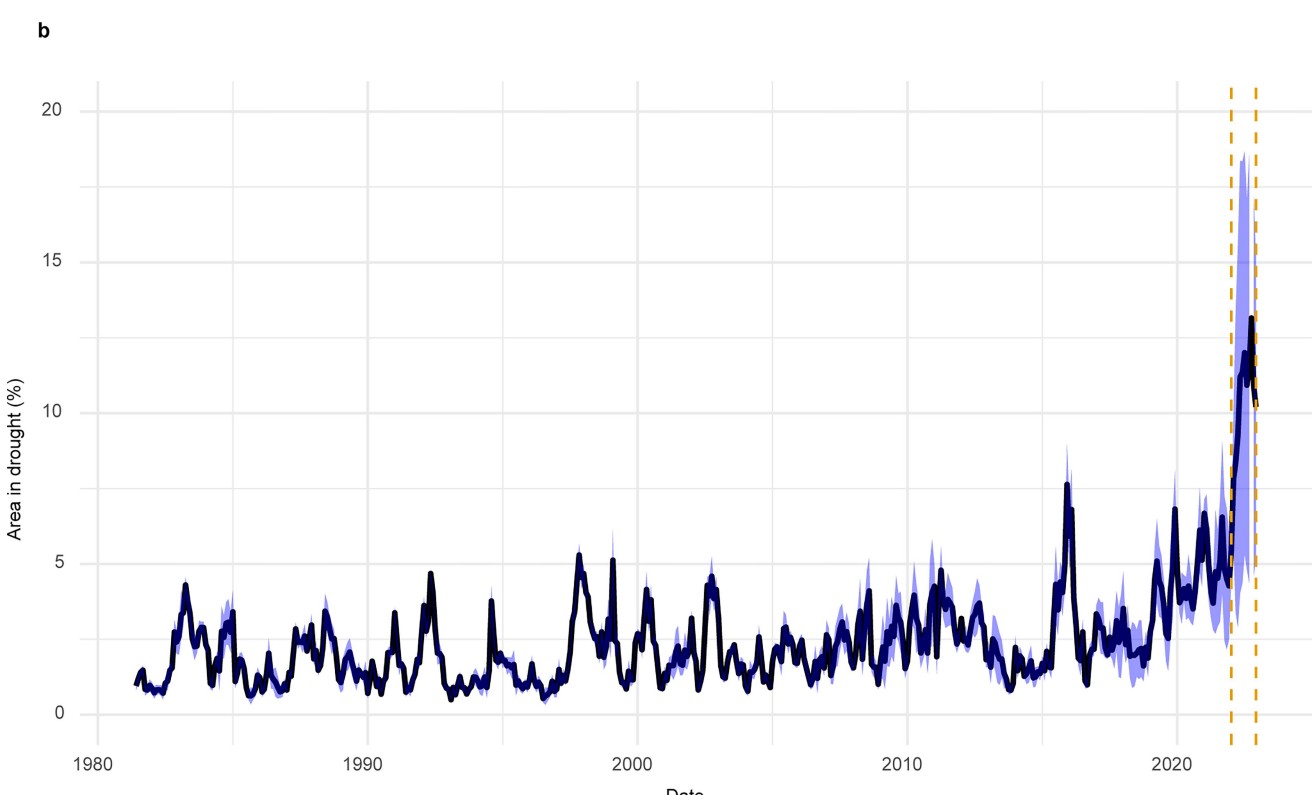

**Extended Data Fig. 4 | Percentage of areas impacted by severe and extreme droughts.** Panels a) and b) show the time series of the percentage of areas affected by severe (SPEI < −1.4) and extreme (SPEI < −1.8) droughts, respectively. The dashed vertical lines mark the last five years (2018–2022), highlighting the increase in drought-affected areas compared to 1981–2017.

**a**

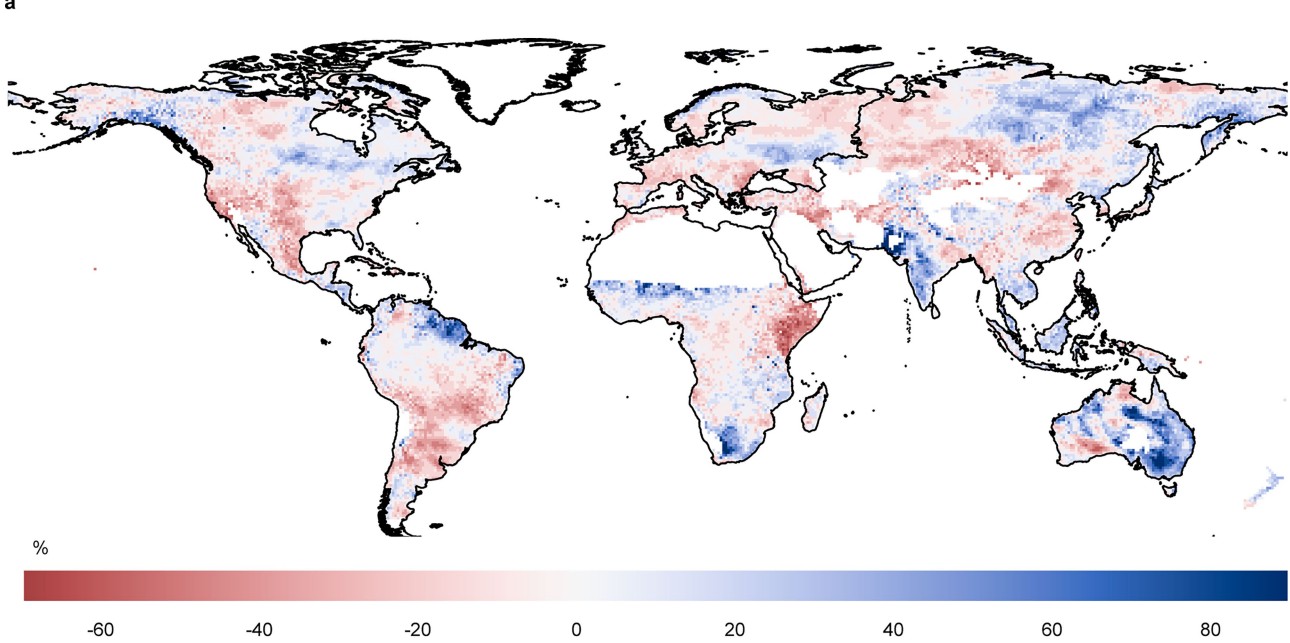

%

**b**

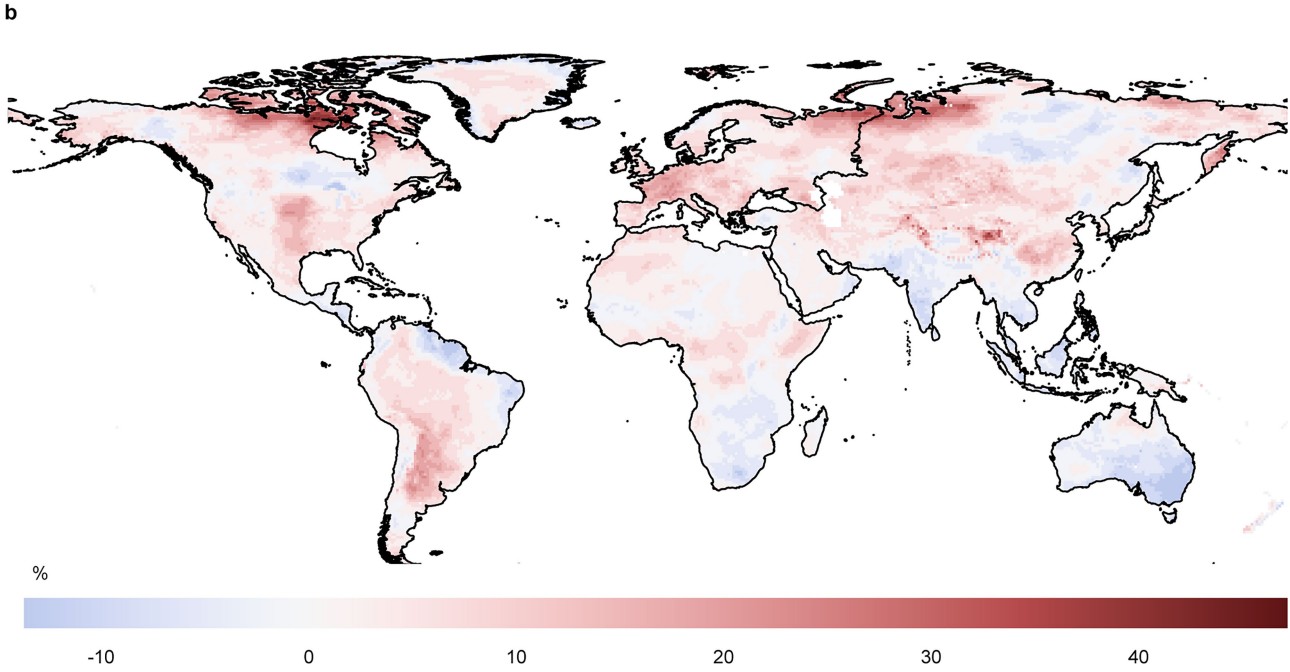

%

**Extended Data Fig. 5 | Annual percentage precipitation and AED anomalies in 2022.** Panels a) and b) show the 2022 precipitation and AED percentage anomalies relative to the long-term mean (1981–2022). Negative values indicate reductions, while positive values indicate increases compared to the long-term average. Precipitation is based on MSWEP, and AED is based on hPET.

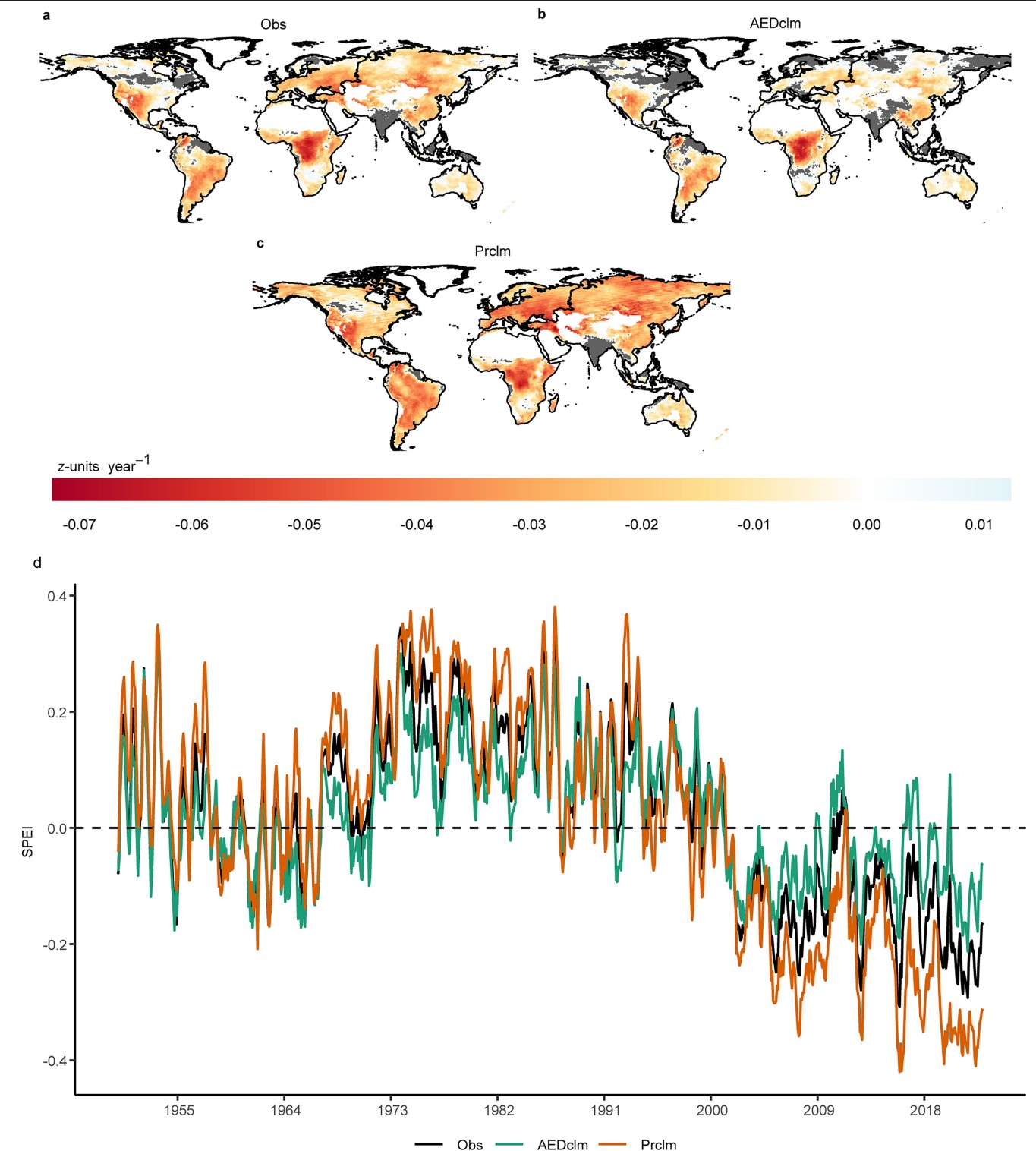

**Extended Data Fig. 6 | Trends in 6-month SPEI based on ERA5 during 1981–2022.** The 6-month SPEI values were computed using the ERA5 meteorological dataset with combinations of observed AED, observed precipitation, climatological AED (AEDclm), and climatological precipitation (Prclm). Panel a) shows the trend based on observed precipitation and AED (Obs). Panel b) presents the trend based on AEDclm and observed precipitation (AEDclm), while panel c) illustrates the trend based on Prclm and observed AED (Prclm). Non-significant trends (P-value > 0.05) are marked in gray to improve clarity. The trends also exclude dryland areas with average annual rainfall below 180 mm. Panel d) displays the quasi-global average (50°S–50°N) 6-month SPEI time series for 1950–2022, based on Obs, AEDclm, and Prclm.

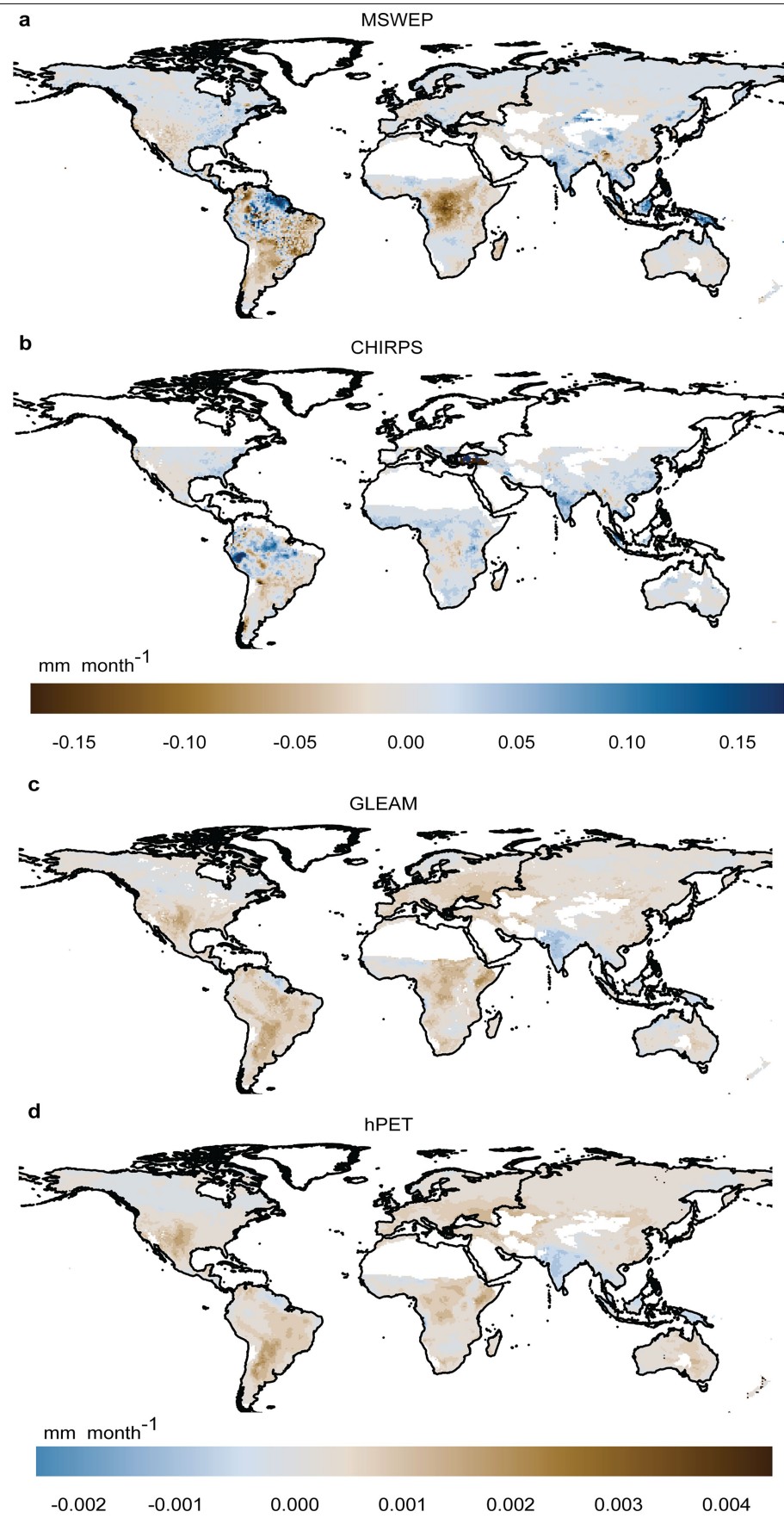

**Extended Data Fig. 7 | Monthly trends in precipitation and AED during 1981–2022.** Panels a) and b) illustrate the trends in monthly precipitation based on monthly MSWEP and CHIRPS datasets, respectively, while panels c) and d) present the trends in monthly AED, derived from GLEAM and hPET datasets, respectively. Note that the CHIRPS dataset covers latitudes up to 50°N.

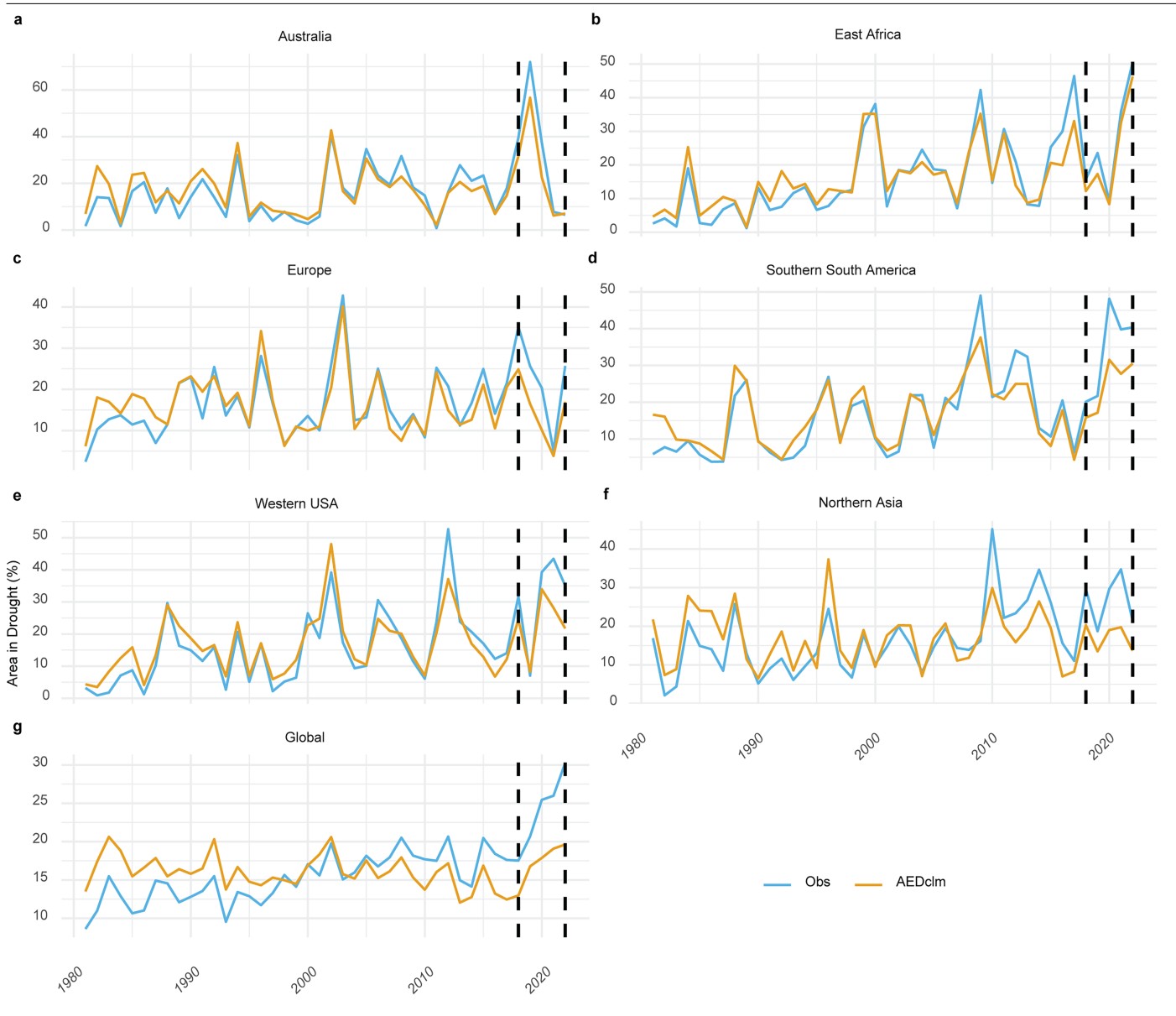

**h**

| Region | Area in drought (%) based on Obs | | Area in drought (%) based on AEDclm | |
|---|---|---|---|---|
| | 1981-2017 | 2018-2022 | 1981-2017 | 2018-2022 |
| Australia | 13.2 | 28.8 | 14 | 19 |
| Southern South America | 10.5 | 27.5 | 10.9 | 17.6 |
| Western USA | 11.6 | 28 | 12 | 19 |
| East Africa | 12 | 21 | 12.5 | 11.3 |
| Northern Asia | 15.3 | 27.6 | 16.5 | 16.6 |
| Europe | 26 | 41 | 27.7 | 26 |
| Global | 15.5 | 27 | 15.7 | 17 |

**Extended Data Fig. 8** | See next page for caption.

**Extended Data Fig. 8 | Percentage of areas affected by drought during 1981–2022.** The time series shows the annual percentage of areas in drought (SPEI < −1) for a) Australia, b) East Africa, c) Europe, d) Southern South America, e) Western USA, f) Northern Asia, and g) globally. The blue lines represent the percentage of areas in drought based on the 6-month SPEI calculated using observed AED and precipitation (Obs), while the orange lines indicate the percentage of areas in drought based on the SPEI computed using observed precipitation and climatological AED (AEDclm). The dashed black vertical lines highlight the period from 2018 to 2022. Panel h) summarizes the percentage of areas affected by drought during 2018–2022, compared to the period 1981–2017, based on both Obs and AEDclm.

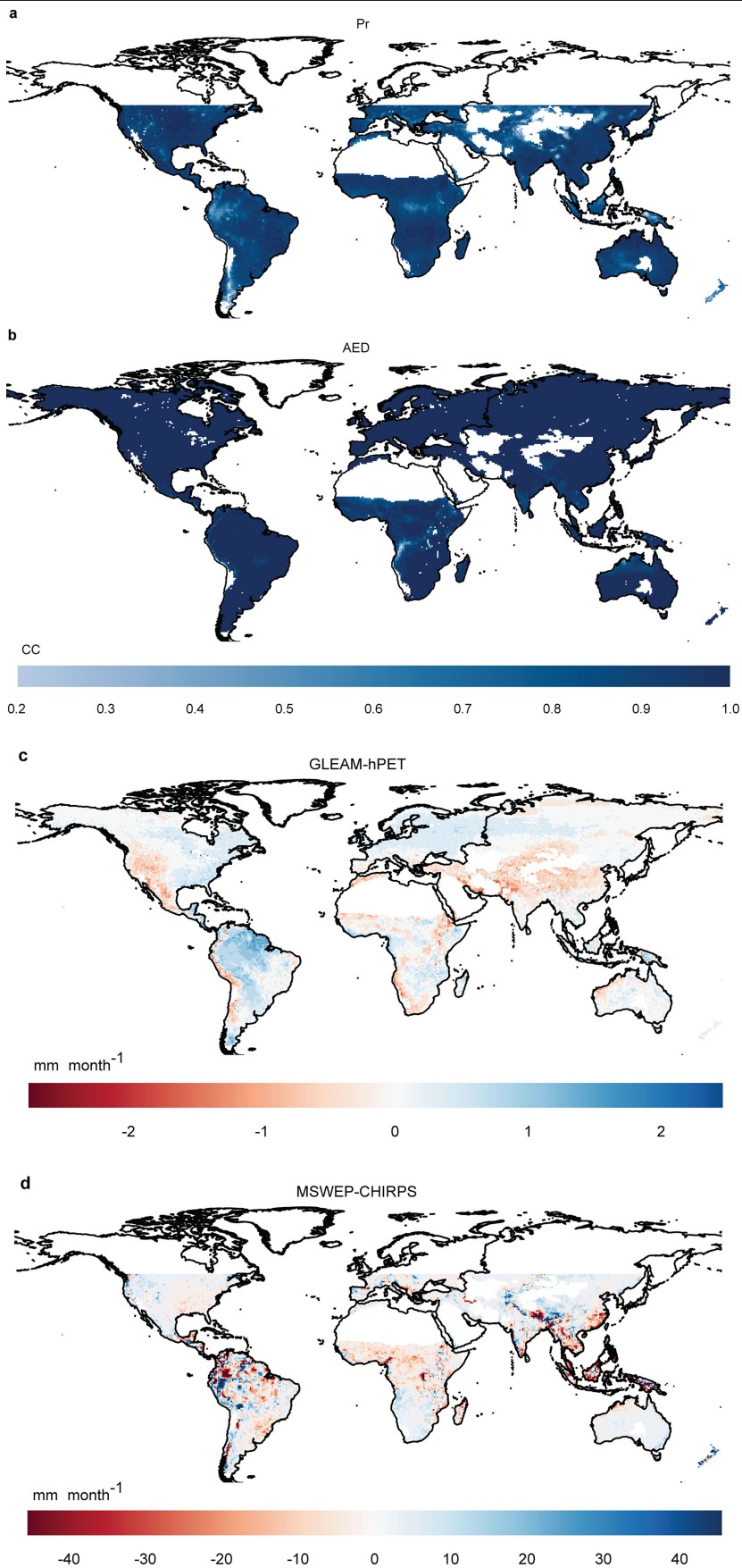

**Extended Data Fig. 9** | See next page for caption.

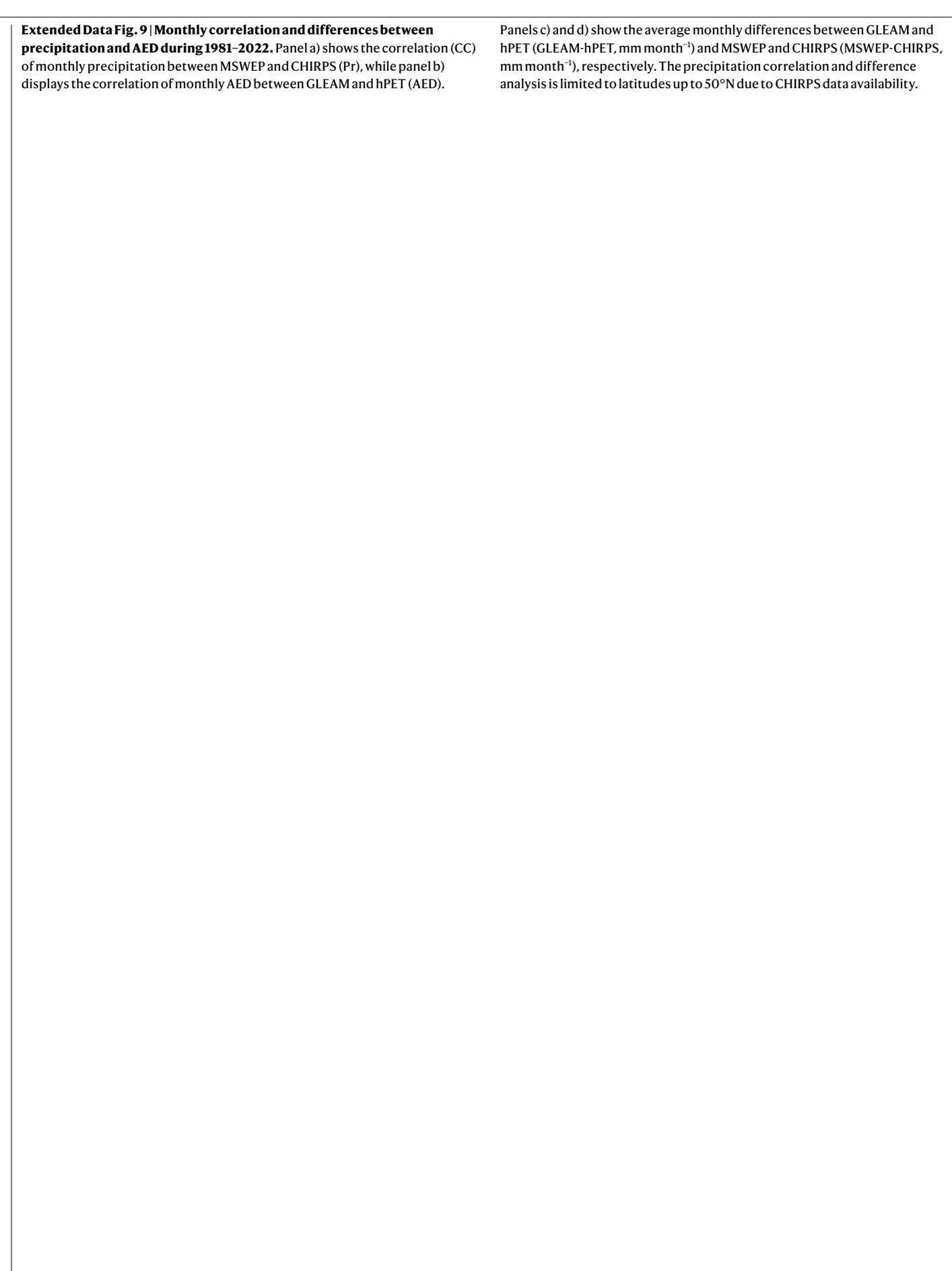

**Extended Data Fig. 9 | Monthly correlation and differences between precipitation and AED during 1981–2022.** Panel a) shows the correlation (CC) of monthly precipitation between MSWEP and CHIRPS (Pr), while panel b) displays the correlation of monthly AED between GLEAM and hPET (AED). Panels c) and d) show the average monthly differences between GLEAM and hPET (GLEAM-hPET, mm month$^{-1}$) and MSWEP and CHIRPS (MSWEP-CHIRPS, mm month$^{-1}$), respectively. The precipitation correlation and difference analysis is limited to latitudes up to 50°N due to CHIRPS data availability.

**Extended Data Table 1 | Categories of wet and dry events**

| SPEI categories | SPEI values[49] |
|---|---|
| Extremely wet | >1.83 |
| Very wet | 1.43 to 1.82 |
| Moderate wet | 1.0 to 1.42 |
| Near Normal | -0.99 to 0.99 |
| Moderately dry | -1.0 to -1.42 |
| Severely dry | -1.43 to -1.82 |
| Extremely dry | < -1.83 |