## [Peer Review file · Nature]

Warming Accelerates Global Drought Severity

Corresponding Author: Dr Solomon Gebrechorkos

Version 1:

Reviewer comments:

Referee #1

(Remarks to the Author)

Thank you for the opportunity to review this interesting paper. I think it might have the potential to be a significant contribution to the literature, and I appreciate the development of the freely available high-resolution SPEI datasets. I believe that with a few manageable revisions to increase the clarity of the argument and a few more caveats it should be suitable for publication.

This paper, while well-written, was at times confusingly presented. I spent more time than I wanted to attempting to understand the following things (which I hope will be easy to fix):

- How is the drought severity time series (SPEI<-1) defined? It says in the methods that it's the # of occurrences of SPEI <-1 but the legend says z units/year. How does it differ from the "drought intensity" discussed in the abstract? Why does Figure 2c, which indicates a large decrease in z-score in Africa and South America when AED is held constant (ie, presumably driven by precipitation deficits), look so different from Figure 3b, which shows a fairly uniform trend difference based on AEDclim?

- I was very confused by Figures 3b and c- are these (trend in HRSPEI)-(trend in AEDCclim) and (trend in HRSPEI) – (trend in Prclim)? It appears that holding AED constant at its climatological value decreases the SPEI, which is not what Figure 3a indicates. If you mean "makes the trend less negative" or "reduces the magnitude of the trend" then it would help your readers to make this much clearer.

- Figure 4: It's not clear how "contributions to X percent of the trend" were calculated. The way this is presented, it appears to argue that precipitation contributed positively to the (negative) SPEI trend, which seems to be the opposite of what you're arguing (and what's shown in Figure 2a). And why is a diverging colormap used to indicate percentages?

A few caveats are, I think, necessary as well. I have no strong desire to re-litigate the endless arguments about atmospheric-based drought indices (PDSI, SPI, SPEI etc) and indices like soil moisture or P-E that account for plant physiological changes. But the neat conclusion of this paper- that changes to demand may lead to changes in drought even in the absence of changes to precipitation deficits – is reliant on the neat index it uses, which takes into account only moisture supply and atmospheric demand. We know that global mean rainfall is increasing quite slowly (at least compared to Clausius-Clapeyron and extreme rain); we also know that PET generally increases as the temperature does (although the difference between daily max and min temperatures obviously matters). Thus, these results are not necessarily surprising, although the confirmation presented is useful. I suppose what I'd like to see is acknowledgement- perhaps in the paragraph beginning on L87- that this doesn't solve the problem of drought analyses being highly metric dependent, and perhaps in the conclusion a little more discussion and caveating of what SPEI changes might mean for drought impacts as actually experienced.

Finally, I found myself unconvinced by the implication that the 2022 drought is attributable to GHG-induced warming. I believe this is likely true, but the attribution statements need to be couched more carefully or based on more rigorous statistical analyses. This is probably fixable by noting 2022 as a particularly dry year and keeping the analysis of AED – I think the section beginning on L247 is fine- but amending the conclusions. I don't think you've shown that this trend has accelerated due to radiative forcing by anthropogenic greenhouse gases (aerosols and natural variability almost certainly played a role here), and I don't think you've explained the intensification post-2017. A formal attribution analysis is likely out of scope for this paper and not required for publication in my opinion, but I'd like to see a little more care taken with these statements.

Minor comments:

- L34 "drought intensity" -> "a particular metric of drought intensity" or similar

- L36 Did global warming accelerate post-2000? I don't see evidence of this. In fact, the literature from around a decade ago is full of papers attempting to explain a "hiatus" in global warming from ~1998-2013.
- L57 "severity of droughts" add "in some regions"?
- L59: pronounced in, but not exclusive to
- L108 it would be useful to have a very brief discussion (probably in Methods) of the differences between MSWEP and CHIRPS / GLEAM vs hPET from ERA5-land. Right now, for example, you just state that both MSWEP and CHIRPs are a combination of satellite, in situ, and modeling, but it would be useful to summarize the areas of disagreement between the datasets that motivate your use of multiple datasets.
- Fig 2 e f using the same colorscales and palette as a-d makes it hard to see the differences.
- L210 this is an optional aesthetic preference, but I think the figure caption is too long and the monthly mean precip and AED discussion should be in the main text
- L236: "lowered the negative trend" it seems to have made the negative trend more positive, please clarify, especially since you use "decreasing the SPEI trend" to mean the opposite above
- L269 "accelerated due to radiative forcing by GHGs" see final comment above. GHG forcing has not accelerated; the rate of increase of global emissions has slowed in recent years.
-

Referee #2

(Remarks to the Author)

Overall Comments

The paper is well written and the reader is able to understand its scope, moreover the results presented are in line with the main motivation behind the study.

In general, this is a fairly good paper, but I have some doubts about its novelty. The authors use a comprehensive set of different inputs to compute the indicators, but the conclusions are mostly known from different studies, including some by co-authors and some properly cited in the introduction.

I am asking myself if this paper is worth a publication in Nature, instead of a climate-based journal, with more specific audience and maybe lower bar regarding the quality/novelty of results.

The use of data is solid, but some key points need to be clarified (see detailed comments below), in particular regarding baseline to compute the SPEI and the use of different AED for specific datasets.

I also feel that the tile should be modified, as in the current version of the manuscript, not enough focus on the 2022 is present.

My overall verdict is that this paper needs minor revisions regarding the methodology, but major revisions in how it is presented to the readers. Maybe it would better fit a different journal, as the results presented may not be enough to justify a publication on Nature portfolio of journals.

Detailed review.

Abstract

I suggest moving the statements regarding the uncertainties in previous global drought assessments to the introduction.

Introduction

The presentation of the topic is well written, though the authors may want to expand the debate over AED, not only citing the papers (as they properly do), but also adding more reasoning and maybe comparisons between different opinions/papers on AED.

If you choose those two datasets for precipitation, I would expect more robust motivation than just a few papers, also some statistics.

It is not immediately clear - without reading the reference - which are the four datasets part of the HRSPEI data. Moreover, it is not clear how you are going to weight or divide the single datasets to combine them in overall results. I am sure that such details will come later in the paper, but a simple line with introductory methodology on this aspect could help.

Trends in..

What is the baseline period used to compute SPEI for the various datasets? Are they homogeneous on this aspect? This is a crucial aspect.

Why cutting lands over 50°N when averaging trends? Is this due to lack of reliable data at high Latitudes? Or because some datasets do not have data there (which is a relevant limitation)? You should motivate such decisions in the core text, not only in the figure captions.

I would suggest including the data on trends/year in a table for a better reading.

You said that "in 2022, annual precipitation across Europe dropped by up to 35% below average". Averaged over which period?

2022 was an exceptional year for drought, as you clearly mention it in the tile, I would like to see a more detailed discussion on its drought characteristics whenever you mention it, not in the extended data, but directly in the core text.

Drivers of..

Why not comparing SPI and SPEI, other than using AEDclm and AED?

Through the text, it is not fully clear which AED method applies to the various SPEI deriving from different datasets. The discussion on Figure 4 is too short, I feel this part has relevant potential for the readers.

Acceleration of..

Also here, as in the most parts of the core paper, the key results could be grouped in a Table, this may improve the reading. Again, the record-breaking 2022 is even included in the title. I want to read more about it, at least I would like to see a dedicated figure-

Conclusions

They are a summary of your findings; I feel that you could expand it, also, including why this study could be of a higher relevance compared to precious global drought assessments.

Methods

Which baseline did you use to compute the SPEI over different periods?

The choice of thresholds for categories of SPEI is a bit arbitrary, could you motivate why using exactly those Numbers instead of the classic -1, -2, etc.?

If you use different methods to compute PET, one might questions if this affects the results. Why not using both methods for all the SPEI computed? Is this possible or one or more datasets lack input data to do that?

Figures and Tables

Captions are very long, consider reducing.

Figure 3 - I would show non-significant trends, maybe with dashed lines, instead of setting them to 0.

Figure 4 is the one of the most interesting parts of the paper, but panels a) and b) are useless, if they sum to 100%, just use a single colour scale focusing on one of them. Same applies to the histograms.

Version 2:

Reviewer comments:

Referee #1

(Remarks to the Author)

Thank you to the authors for such a through and attentive revision- I very much appreciate their hard work. I have only one small point I'd like to see the authors address before publication.

I continue to be baffled by the claim that global warming began to accelerate in 2000, and that this made the influence of AED more pronounced. This does not appear to be supported by anything in the analysis, the response to reviewers, or the literature. (Lenssen et al 2019 makes no statement about post-2000 acceleration, and to my knowledge none of the global temperature datasets show a step change in trends at this time). Moreover, I don't see that Figure 3a shows "the impact of AED has become increasingly evident since 2000", as claimed on line 257. What I see is a large trend in an hPET_Prclm time series that passes through zero circa 2000 because that's the middle of the SPEI baseline period. This claim, which appears in the abstract and throughout the text, is a strong attribution statement and must be supported by strong evidence. My prior- subject to updating, of course- is that it's not true, and that high-frequency temporal variations on top of the long-term trends in global drought behavior are more readily explained by SST patterns, dynamics, and other factors. I suggest eliminating the "post-2000" claim and discussion of global warming acceleration or presenting quite a bit more evidence and rigorous attribution to justify it. It's fine to note the dramatic increase in drought are in the last five years- I think explaining the underlying drivers is probably out of scope here- and the discussion of the 2022 drought is now OK.

Referee #2

(Remarks to the Author)

I carefully read the new version of the manuscript and all the point-by-point replies by authors to the (I must admit) large number of suggestions made by the reviewers.

I am positively surprised by the new version of the manuscript, the authors crafted robust and detailed responses and the paper is now ready, in my opinion for publication.

Reply to Referee #1

We appreciate the reviewer for dedicating their time and providing valuable and detailed feedback. We have addressed all comments as follows.

Update: The AED from the GLEAM model has been updated to the latest version (version 4.2a), which is based on the more comprehensive Penman equation, replacing the previously used Priestley-Taylor equation. This update aims to provide the most accurate analysis of droughts. As a result, some values have changed, as you will notice in the paper, but the update has not altered the study's findings.

Thank you for the opportunity to review this interesting paper. I think it might have the potential to be a significant contribution to the literature, and I appreciate the development of the freely available high-resolution SPEI datasets. I believe that with a few manageable revisions to increase the clarity of the argument and a few more caveats it should be suitable for publication.

Authors' response:

- Thank you for recognizing our efforts and the potential need for high-resolution drought datasets, and for providing detailed feedback and recommendations to improve the quality of the paper.

This paper, while well-written, was at times confusingly presented. I spent more time than I wanted to attempting to understand the following things (which I hope will be easy to fix): How is the drought severity time series (SPEI<-1) defined? It says in the methods that it's the # of occurrences of SPEI <-1 but the legend says z units/year. How does it differ from the "drought intensity" discussed in the abstract?

Authors' response:

- For clarity, we have replaced the word "severity" with "magnitude", and use "severity" as an overarching term to refer to all aspects of drought: intensity, magnitude, duration and extent.
- The definition of magnitude is based on the classical approach of the integrated value of the drought index below the drought threshold (Dracup et al. 1980) and is widely used in the literature (Van Loon 2015; Lorenzo-Lacruz et al. 2013; Fleig et al. 2006). In this study, we use $\text{SPEI} < -1$ as the drought threshold, with values between -1 and 1 considered near-normal conditions and values > 1 indicating wet conditions. For each drought event ($\text{SPEI} < -1$), drought magnitude is calculated by summing the monthly SPEI values over its duration, reflecting the event's aggregated value below the threshold. For the annual trend analysis, we sum all SPEI values below -1 for each year and calculate the trend using the Mann-Kendall test. Since magnitude is determined by the sum of SPEI values, it retains the units of SPEI, and the trend is expressed as z-units per year. Drought intensity, on the other hand, refers to the maximum negative value of SPEI observed during the event.
- To improve the clarity of the terminology, we have noted that "severity" refers to the collective of drought metrics (see methodology), and have added clear definitions of

magnitude, intensity, frequency, and duration, with severity serving as an overarching term for all drought metrics in the revised manuscript 470-479 as:

“In this study, we use $SPEI < -1$ as the threshold to define a drought, with values between -1 and 1 considered near-normal conditions and values > 1 indicating wet conditions. (Table 1). Using $SPEI < -1$ values, we assessed key drought metrics: magnitude, duration, intensity, and frequency. We follow the classic approach⁸¹ and widely adopted methods to define these metrics. Drought magnitude is calculated as the cumulative sum (running total) of $SPEI < -1$ values during a drought event. Drought intensity, on the other hand, is defined as the maximum negative value of SPEI observed during the event. Duration represents the run length of consecutive months with $SPEI < -1$, and frequency is the total number of drought events within a given period⁸¹. Finally, severity is used as an overarching term to refer to all aspects of drought: intensity, magnitude, duration and extent.”

Why does Figure 2c, which indicates a large decrease in z-score in Africa and South America when AED is held constant (ie, presumably driven by precipitation deficits), look so different from Figure 3b, which shows a fairly uniform trend difference based on AEDclim?

Authors' response:

- Figures 2c and 3b depict different time scales and metrics, which explains the differences. Figure 2c shows the annual trend in drought magnitude, where AED is held constant (AEDclim), while Figure 3b illustrates the difference in the 6-month SPEI trend between observed AED and AEDclim, not the magnitude. These differences in metrics and time scales lead to the contrasting patterns in the two figures. As you noted, the large decrease in magnitude in Africa and South America in Figure 2c is primarily driven by precipitation deficits, with AED kept constant (AEDclim). To clarify this point further, we have added the following text in lines 216-219:

“In comparison to much of the world, parts of Africa and South America exhibit a greater increase and decrease in drought magnitude and frequency, respectively, highlighting that these trends are primarily driven by precipitation deficits.”

I was very confused by Figures 3b and c- are these (trend in HRSPEI)-(trend in AEDCclim) and (trend in HRSPEI) – (trend in Prclim)? It appears that holding AED constant at its climatological value decreases the SPEI, which is not what Figure 3a indicates. If you mean “makes the trend less negative” or “reduces the magnitude of the trend” then it would help your readers to make this much clearer.

Authors' response:

- Your interpretation is correct, and we agree that the distinction between Figures 3b and 3c can be confusing. Figure 3b shows the difference in trends between SPEI based on observations (observed precipitation and AED) and SPEI based on observed precipitation and the climatology of AED (AEDclim). Figure 3c shows the trend difference between SPEI based on observations (observed precipitation and AED) and SPEI based on observed AED and the climatology of precipitation (Prclim). For clarity, we updated the figure and figure legend as follows:

Fig. 3. Monthly time series and trend differences for 6-month SPEI based on observed and climatological AED (AEDclm) and precipitation (Prclm) during 1981–2022. Panel a) presents the quasi-global average (50°S–50°N) 6-month SPEI based on AEDclm, Prclm, and

HRSPEI. MSWEP_AEDclm and CHIRPS_AEDclm refer to the average SPEI based on MSWEP and CHIRPS precipitation and AEDclm (mean of GLEAM and hPET). GLEAM_Prclm and hPET_Prclm show the average SPEI based on AED from GLEAM and hPET and Prclm (mean of MSWEP and CHIRPS). Panel b) shows the trend difference between SPEI based on observations (observed precipitation and AED) and SPEI based on observed precipitation and the climatology of AED (AEDclm). Figure c) shows the trend difference between SPEI based on observations and SPEI based on observed AED and climatology of precipitation (Prclm). Non-significant trends (P-value > 0.05) are set to zero to enhance clarity. The trend excludes dry land areas with average annual rainfall below 180 mm. For regions above 50°N, the trend is based on the mean of MSWEP_hPET and MSWEP_GLEAM, as CHIRPS is available up to 50°N.

- Figure 3c shows the trend difference between SPEI based on observations (observed precipitation and AED) and SPEI based on observed AED and the climatology of precipitation (Prclm).
- As shown in Figure 3b, the difference between the SPEI based on observed AED and precipitation and AEDclm and observed precipitation is negative, which indicates that the trend based on AED is more negative than the trend based on AEDclm. This suggests that AED contributes to the observed decreasing trend in SPEI (i.e. 'more drought').
- Figure 3a shows the SPEI time series for observed data, AEDclm, and Prclm. Holding AED constant at its climatological value (AEDclm) results in a less negative trend in SPEI compared to the trend based on observed data (HRSPEI). This indicates that observed changes in AED made the observed trend in SPEI more negative. Furthermore, holding Pr constant (Prclm) results in a more negative trend in SPEI indicating that observed increases in precipitation offset observed decreases in SPEI driven by AED increases.
- To make this clearer, as you suggested, we have added some text in the manuscript (lines 249–252) to indicate that holding AED to its climatology (AEDclm) switch the trend from negative to positive:

“The quasi-global average 6-month SPEI trend, based on observed precipitation and AEDclm, is 0.002 year^{-1} , which is about 131% higher than the observed trend (**Fig. 3**), indicating that holding AED to its climatological value results in a positive trend...”

Figure 4: It's not clear how “contributions to X percent of the trend” were calculated. The way this is presented, it appears to argue that precipitation contributed positively to the (negative) SPEI trend, which seems to be the opposite of what you're arguing (and what's shown in Figure 2a). And why is a diverging colormap used to indicate percentages?

Authors' response:

- To clarify, the "contributions to X percent of the trend" are calculated by comparing the observed trend in the 6-month SPEI with trends in SPEI based on AEDclm and Prclm. Specifically, the contribution of AED is determined by subtracting the trend in SPEI

based on AEDclm from the observed SPEI trend. Similarly, the contribution of precipitation (Pr) is computed by subtracting the trend based on Prclm from the observed trend. These calculations reflect the relative magnitude of each factor's contribution to the overall trend, without explicitly separating their effects on positive and negative trends. To address this, we have revised the figure legend to clearly explain how the contributions are calculated and to indicate that the values represent the magnitude of contributions without sign differentiation.

- As per your recommendation, we have also replaced the diverging colour map with a continuous one to enhance visual clarity and ease of interpretation as:

Fig. 4: Percentage of contribution (%) of AED and precipitation (Pr) to 6-month SPEI trends. Figures a) show the percentage of contribution of AED and b) show the percentage of contribution of Pr for the observed changes in 6-month HRSPEI during 1981-2022. The

contributions are computed by calculating the difference between the observed trend and the trend based on the climatological values of AED (AEDclm) and precipitation (Prclm). The contribution of AED is determined by the difference between the trend using observed AED and Pr and the trend using observed Pr and AEDclm. Similarly, the contribution of Pr is calculated as the difference between the trend using observed Pr and AED and the trend using observed AED and Prclm. The percentage contribution of each factor is then calculated as the absolute value of the difference divided by the total absolute difference, providing a relative measure of each factor's influence on the observed trend. The lower panel (c) provides the regional and global average contribution of Pr and AED to the changes in 6-month SPEI.

A few caveats are, I think, necessary as well. I have no strong desire to re-litigate the endless arguments about atmospheric-based drought indices (PDSI, SPI, SPEI etc) and indices like soil moisture or P-E that account for plant physiological changes. But the neat conclusion of this paper- that changes to demand may lead to changes in drought even in the absence of changes to precipitation deficits – is reliant on the neat index it uses, which takes into account only moisture supply and atmospheric demand. We know that global mean rainfall is increasing quite slowly (at least compared to Clausius-Clapeyron and extreme rain); we also know that PET generally increases as the temperature does (although the difference between daily max and min temperatures obviously matters). Thus, these results are not necessarily surprising, although the confirmation presented is useful. I suppose what I'd like to see is acknowledgement- perhaps in the paragraph beginning on L87- that this doesn't solve the problem of drought analyses being highly metric dependent, and perhaps in the conclusion a little more discussion and caveating of what SPEI changes might mean for drought impacts as actually experienced.

Authors' response:

- We agree that it is useful to discuss the SPEI index in this context. We have revised the introduction to highlight this, with the following added in lines 93-96:

“Whilst drought can be characterized in many ways to reflect different meteorological, hydrological and ecologically drivers, consideration of the influence of AED with respect to precipitation is crucial to understand how climate change is impacting on changes in drought.”

- Additionally, we expanded the discussion section to discuss the focus of this study and to highlight the importance of considering other variables such as soil moisture and vegetation conditions, especially to better understand the actual impacts of drought as experienced on the ground. For example, the following is added to the manuscript, **lines 406-413**, as recommended:

“The observed increase in drought severity aligns with associated impacts on agricultural, environmental, and hydrological systems, as seen in events like the 2022 European drought, which contributed to enhanced tree mortality^{70,71}, increased forest fires⁷², and long-term soil moisture decline⁶⁰. Although the SPEI is a meteorological drought index that focuses on the influence of precipitation and AED, it can also represent agricultural, water resources and ecosystem drought related impacts effectively⁷³⁻⁷⁵. However, further studies are needed, considering variables such as soil moisture, vegetation stress, and hydrological flows for better understanding the broader impacts of the observed changes on ecosystems and human activities.

Finally, I found myself unconvinced by the implication that the 2022 drought is attributable to GHG-induced warming. I believe this is likely true, but the attribution statements need to be couched more carefully or based on more rigorous statistical analyses. This is probably fixable by noting 2022 as a particularly dry year and keeping the analysis of AED – I think the section beginning on L247 is fine- but amending the conclusions. I don't think you've shown that this trend has accelerated due to radiative forcing by anthropogenic greenhouse gases (aerosols and natural variability almost certainly played a role here), and I don't think you've explained the intensification post-2017. A formal attribution analysis is likely out of scope for this paper and not required for publication in my opinion, but I'd like to see a little more care taken with these statements.

Authors' response:

- We agree with the reviewer on the existing difficulties and uncertainties in assessing the potential contribution of anthropogenic forcing to the severity of a particular drought. There are several issues affecting this assessment, which include data availability, length of series, uncertainty in model simulations, etc. Nevertheless, there is recent new evidence that suggests that drought in 2022, at least in Europe, was strongly affected by anthropogenic climate change, which enhanced atmospheric evaporative demand (AED), affecting the severity of drought (Bevacqua et al. 2024). Two recent studies have been published considering the hydrological implications of the enhanced AED in the year 2022 in Europe (Bevacqua et al. 2024a; Garrido-Perez et al. 2024a). In fact, the study by Garrido-Perez et al. (2024) provides a long-term perspective of the event, demonstrating that the strong decline of streamflow cannot be directly explained by the precipitation deficit (which could be linked to natural variability associated to circulation mechanisms) but mostly to the enhanced AED, which would reinforce hydrological drought severity given the large water losses by evaporation. The study by Bevacqua et al. (2024) goes in the same direction, indicating that the temperature anomaly was the main driver of drought severity in most of Europe, directly related to human-induced climate change. We acknowledge that, in regions where no formal attribution studies have been conducted, it is challenging to assess the precise role of anthropogenic climate change in the 2022 drought. While we recognize the contribution of human-induced climate change to the intensification of drought conditions, we also understand that the broader global implications—particularly regarding the 2022 drought—are less clear and beyond the scope of this paper.
- In response to this, we have removed the “Record-Breaking 2022 drought” from the title to avoid implying direct attribution to anthropogenic GHG emissions. Also, we have revised the discussion section in order to qualify the statements at the global scale but including the evidence at least for the potential attribution of the drought severity to the human-induced change in Europe (lines 391-400) :

“In Europe, the severity of the 2022 drought event can be largely attributed to anthropogenic global warming, since the strong anomalies observed in streamflow and soil moisture cannot be explained by the precipitation deficit alone, but mostly by enhanced AED, which reinforced large water losses by evaporation^{46,63,64}. Moreover, ecological drought severity recorded in Europe's natural forests cannot be fully explained without considering the influence of high temperatures and AED on plant physiology. In other regions of the world, where no formal attribution studies have been conducted, it is challenging to robustly assess the potential role of human-induced climate change in the severity of the 2022 drought. Nevertheless, given the

attribution of enhanced AED on the severity of the 2022 drought in Europe, and the increase in severity globally driven by enhanced AED as shown in this study, it is reasonable to conclude that anthropogenic global warming likely contributed to exacerbate global drought severity in 2022.”

Minor comments:

L34 “drought intensity” -> “a particular metric of drought intensity” or similar

Authors’ response:

- Thank you for your comment. We agree that specifying the metric is crucial for clarity. As noted above, drought intensity is defined as the maximum negative value of SPEI observed during an event. We have added clear definitions of all drought metrics (intensity, frequency, magnitude, and severity) in the methodology section (Lines 470–479) to enhance clarity.

L36 Did global warming accelerate post-2000? I don’t see evidence of this. In fact, the literature from around a decade ago is full of papers attempting to explain a “hiatus” in global warming from ~1998-2013.

Authors’ response:

- Thank you for your comment. We have used NASA’s “Global Annual Mean Surface Air Temperature Change” dataset (https://data.giss.nasa.gov/gistemp/graphs_v4/), which shows a rise in terrestrial air temperatures of approximately 1°C from the late 1990s to 2022. This is also highlighted by Lenssen et al. (2019), as presented in “Improvements in the GISTEMP Uncertainty Model,” which includes a detailed time series chart. Additionally, based on ERA5, the trend in SPEI is positive before 1990 and negative after 1990. The role of AED is clearly visible when precipitation is held to its climatological value. For instance, based on ERA5 and HRSPEI, the SPEI values are negative post-2000 when precipitation is fixed to its climatological value (Prclm), indicating that the changes are driven by variations in AED.
- We have added a few lines in the section "Drivers of Changes in Drought," referencing Lenssen et al. (2019) to provide further clarity and support as:

Lines: 257-263: “These findings indicate that AED changes from 1981 to 2022 intensified both the downward trend in SPEI and the expansion of drought-affected areas, especially after 2000, which can be attributed to the increase in terrestrial air temperature from late 1990s to 2022⁴². Before 2000, the time series based on Prclm shows positive SPEI values, becoming negative after 2000 (Fig. 3a). Additionally, ERA5 data shows a positive trend from 1960–1990, shifting to negative from 1991–2022 (Extended Data Fig. 7d), highlighting the increased impact of AED as precipitation remains fixed at its climatological value. ”

L57 “severity of droughts” add “in some regions”?

Authors’ response:

- We have edited the sentence to include "in some regions" to clarify that the severity of droughts is projected to increase in specific areas.
Lines 68-69: "Future projections from climate models also suggest a heightened severity of droughts in some regions due to decreased precipitation and enhanced AED"

L59: pronounced in, but not exclusive to

Authors' response:

- We have now edited the entire paragraph for clarity.

L108 it would be useful to have a very brief discussion (probably in Methods) of the differences between MSWEP and CHIRPS / GLEAM vs hPET from ERA5-land. Right now, for example, you just state that both MSWEP and CHIRPs are a combination of satellite, in situ, and modeling, but it would be useful to summarize the areas of disagreement between the datasets that motivate your use of multiple datasets.

Authors' response:

- Thank you for this valuable suggestion. We have added a description in the Methods section to discuss the differences in underlying methods between MSWEP and CHIRPS, as well as GLEAM and hPET. We have also updated the AED from the GLEAM model to the latest version (version 4.2a), which is based on the more comprehensive Penman equation, replacing the previously used Priestley-Taylor equation.
- We also highlight the differences between these data sets. For example, CHIRPS uses only geostationary thermal infrared observations, whilst MSWEP also uses microwave observations. Convergence between the two products helps reduce concerns about the uncertainties due to different approaches and changes in the constellation of Earth-observing satellites that can affect the robustness of their representation of changes over time.
- We have also added a correlation map between the MSWEP and CHIRPS and GLEAM and hPET and the difference between the MSWEP and CHIRPS and GLEAM and hPET in Extended Data Fig. 9.
- For MSWEP and CHIRPS, the following has been added in lines 489-521 with a monthly correlation map and average monthly difference in Extended Data Fig. 9.

"CHIRPS and MSWEP were chosen as they generally out-perform other similar gridded precipitation datasets when compared to ground observations⁵²⁵³. CHIRPS (0.05°) is particularly designed for monitoring droughts and detecting environmental changes, providing daily precipitation estimates from 1981 to present. It combines satellite-derived Climate Hazards Center Infrared Precipitation (CHIRP) and the Climate Hazards Group Precipitation Climatology (CHPclim) with ground station data from the Global Historical Climate Network (GHCN) and many other sources. The CHIRPS product benefits from a high degree of homogeneity, provided by its simple but consistent foundation of geostationary thermal infrared satellite observations. CHIRPS also incorporates unique observation inputs from Africa, Latin America and Central America. MSWEP (0.1°) has been designed with both accuracy and homogeneity in mind, providing 3-hourly precipitation estimates from 1979 to present. It integrates daily observations from over 77,000 stations from various national and

international data sources, satellite estimates from infrared- and microwave-based satellite datasets, and reanalysis data, offering accurate global precipitation data from 1979 to present. Both CHIRPS and MSWEP have previously been evaluated globally using statistical metrics such as Kling–Gupta Efficiency (KGE) and Nash-Sutcliffe Efficiency (NSE), as well as various bias and error metrics^{52,53}. For instance, MSWEP outperformed 22 other global precipitation datasets in capturing daily precipitation from 76,086 gauging stations and in driving hydrological models across 9,053 catchments⁵². Additionally, both MSWEP and CHIRPS were found to outperform other high-resolution gauge-based datasets in modelling daily, monthly, and annual streamflow across 1,825 streamflow gauges⁵³. However, both datasets remain subject to inherent uncertainties, and therefore, considering both helps reduce biases and obtain more reliable estimates, given that they are somewhat independent. For example, they differ in their data sources with CHIRPS using only geostationary thermal infrared observations, whilst MSWEP also uses microwave observations, and they use different sets of station data to correct locally. Despite these differences, the monthly correlation between MSWEP and CHIRPS shows a high correlation across most regions, except for Central Asia (Extended Data Fig. 9a). The average monthly difference between the two datasets varies spatially, reaching up to ± 40 mm (Extended Data Fig. 9c). Notably, larger discrepancies occur in regions such as the Amazon, Central Africa, and parts of Southeast Asia. Such convergence between the two products helps reduce concerns about the uncertainties due to different approaches and changes in the constellation of Earth-observing satellites that can affect the robustness of their representation of changes over time.

- For GLEAM and hPET, as we updated GLEAM to its latest version (v4.2a), it now utilizes the most comprehensive Penman approach, similar to hPET, which is based on the FAO-56 Penman-Monteith equation. Both methods incorporate similar physical principles, including considerations for temperature, radiation, humidity, and wind speed. However, GLEAM explicitly computes aerodynamic resistance. These differences, alongside their methodological alignment, justify the complementary use of GLEAM and hPET in this study to provide a robust estimate of AED. The following discussion is added in the Methods section (lines: 529–537) with a correlation map between GLEAM and hPET, along with their monthly difference map provided in Extended Data Fig. 9b.

“hPET is based on the FAO Penman-Monteith equation, which computes reference crop evaporation by assuming certain surface and aerodynamic characteristics that are constant in time. In contrast, GLEAM4 calculates aerodynamic conductance as a dynamic variable depending on ecosystem characteristics and local meteorology and therefore is space and time dependent. Nonetheless, given the dominant influence of radiative forcing and atmospheric aridity in both computations, their estimates are overall similar. The monthly average difference between GLEAM and hPET is up to ± 3 mm (Extended Data Fig. 9d), and their correlation exceeds 0.9 across 91% of the global land surface (Extended Data Fig. 9b).”

Extended Data Fig. 9: Monthly correlation between CHIRPS and MSWEP for precipitation, and GLEAM and hPET for AED, along with their differences during 1981–2022. The precipitation correlation is limited to latitudes up to 50°N due to the availability of CHIRPS data only up to this latitude. Panel a) shows the correlation between MSWEP and CHIRPS, while panel b) displays the correlation between GLEAM and hPET AED. Panels c) and d) show the average monthly difference between MSWEP and CHIRPS, and GLEAM and hPET, respectively.

Fig 2 e f using the same color scales and palette as a-d makes it hard to see the differences.

Authors' response:

- We have revised the colour scales for Figures 2e and 2f to improve clarity and distinguish them from Figures 2a–d. Additionally, we have also updated the figure legend to explain the computation of magnitude and frequency:
Lines 184-186: “Magnitude is calculated as the cumulative sum of $\text{SPEI} < -1$ values during a drought event for each year, while frequency represents the number of events in a year with $\text{SPEI} < -1$.”

Fig. 2. Trends in drought magnitude and frequency for 6-month SPEI based on observed and AEDclm during 1981–2022. Figure a) shows the trend in magnitude (z-units year⁻¹) and b) the frequency (months year⁻¹) of droughts (SPEI < -1) for the period 1981–2022 based on MSWEP_hPET. The trend in magnitude and frequency based on observed precipitation and AEDclm is displayed in c) and d), respectively. Figures e) and f) show the difference in trend between the observed data and AEDclm for drought magnitude and frequency, respectively. Non-significant trends (P-value > 0.05) are set to zero for clarity. Magnitude is calculated as

the cumulative sum of SPEI < -1 values during a drought event for each year, while frequency represents the number of events in a year with SPEI < -1. The lower time series panels show the regional annual average magnitude (g-i) and annual frequency (m-r) of droughts averaged over South America (g and m), Africa (h and n), Australia (i and o), Europe (j and p), Asia (k and q), and North America (l and r). The trend and regional average exclude dry land areas with average annual rainfall below 180 mm.

L210 this is an optional aesthetic preference, but I think the figure caption is too long and the monthly mean precip and AED discussion should be in the main text

Authors' response:

- Thank you. We have shortened the figure caption for clarity and moved the discussion of precipitation and AED to the methodology section, as recommended.

L236: "lowered the negative trend" it seems to have made the negative trend more positive, please clarify, especially since you use "decreasing the SPEI trend" to mean the opposite above

Authors' response:

- Thank you for pointing this out. We agree that the wording might cause confusion, so we have removed it for clarity. What we intended to convey is that the negative SPEI trend is more pronounced when using climatological precipitation compared to observed precipitation. This indicates that the world would appear drier if precipitation remained constant at climatological levels. By contrast, when observed precipitation is used, the SPEI trend is less negative, as shown in Figure 3c. Now it reads as:

Lines 315-324: "Overall, even though precipitation accounts for 60% of the global average SPEI trend during 1981–2022, the role of AED, contributing 40%, is substantial (Fig. 4). This is especially notable considering the stronger sensitivity of SPEI to precipitation than to AED in most land regions⁵⁷. Furthermore, although the trend contribution covers 1981–2022, the impact of AED has become increasingly evident since 2000 (Fig. 3a)."

L269 "accelerated due to radiative forcing by GHGs" see final comment above. GHG forcing has not accelerated; the rate of increase of global emissions has slowed in recent years.

Authors' response:

- We have removed this part, as our study does not specifically examine the impact of radiative forcing by GHGs in detail and we have updated the paragraph as:

Lines 375-379: "According to the Standardized Precipitation Evapotranspiration Index (SPEI), over the past 42 years (1981–2022), global drought severity has intensified. In the last 5–10 years, this trend has accelerated as a consequence of the strong increase in AED, which is directly related to global warming and increased vapour pressure deficit¹⁸, as the water supply to the atmosphere is not enough to compensate for the large temperature increase."

Reply to Referee #2

We would like to express our sincere appreciation to the reviewer for dedicating their time and providing valuable, detailed feedback on our manuscript. We have made the necessary revisions to address each point as detailed below.

Overall Comments

The paper is well written and the reader is able to understand its scope, moreover the results presented are in line with the main motivation behind the study. In general, this is a fairly good paper, but I have some doubts about its novelty. The authors use a comprehensive set of different inputs to compute the indicators, but the conclusions are mostly known from different studies, including some by co-authors and some properly cited in the introduction. I am asking myself if this paper is worth a publication in Nature, instead of a climate-based journal, with more specific audience and maybe lower bar regarding the quality/novelty of results. The use of data is solid, but some key points need to be clarified (see detailed comments below), in particular regarding baseline to compute the SPEI and the use of different AED for specific datasets. I also feel that the title should be modified, as in the current version of the manuscript, not enough focus on the 2022 is present. My overall verdict is that this paper needs minor revisions regarding the methodology, but major revisions in how it is presented to the readers. Maybe it would better fit a different journal, as the results presented may not be enough to justify a publication on Nature portfolio of journals.

Authors' response:

- Thank you for your thoughtful feedback and for recognizing the quality of our paper. We greatly appreciate your suggestions for improving its clarity and presentation. Following your recommendation, we removed the “and Record-Breaking 2022 Drought” from the title to reflect the focus of the paper on long-term changes.
- We believe that this research addresses critical gaps and uncertainties in the existing literature, particularly as highlighted in the recent IPCC AR6. As pointed out in the introduction, there remain significant uncertainties in drought trend analyses due to issues related to coarse forcing datasets, methodological approaches to compute AED, and the lack of consistency in the use of precipitation datasets. These uncertainties have persisted in the literature, leading to contrasting conclusions. In our study, we take a novel approach by addressing these key sources of uncertainty:
 1. Precipitation datasets: Rather than relying on readily available datasets, we conducted comprehensive evaluations of precipitation datasets on a global scale, identifying the most accurate ones for use in our analysis.
 2. SPEI calculation: While previous studies often used models requiring specific calibration periods, we opted to use the SPEI, which incorporates the entire period of data (1981–2022) to compute the indices. This eliminates the need for

calibration, making our results more consistent and reducing uncertainty compared to methods that are dependent on arbitrary calibration periods, which led to large uncertainties.

3. AED computation: The calculation of AED is a known source of uncertainty, as methods relying solely on temperature or simplified approaches can introduce significant errors. To address this, we use recently derived and well-validated AED datasets, such as hPET and GLEAM, which incorporate a wide range of climatic variables. This complexity ensures that the AED component of our analysis is as accurate as possible. As noted earlier, to further enhance the accuracy of our results, we have updated the AED from the GLEAM model to its most recent version (4.2a), which is based on the comprehensive Penman equation, in contrast to the Priestley-Taylor equation, which considers a more limited set of climate variables.
 4. Moreover, in this study, we quantify the impact of AED on global drought severity, based on an observational dataset, a topic that has been discussed generally in the literature but never rigorously quantified and isolated in this context. Our study fills this important gap by providing robust evidence of the significant role AED plays in explaining recent drought severity trends, which has important implications for anticipating future drought trends, particularly in regions that may be increasingly affected by AED due to global warming.
- To emphasize the novelty of this work, we have revised the introduction section. For instance, the following text has been added;

Lines 144-157: “Numerous studies have analyzed drought trends at the regional and national scales using the SPEI, demonstrating its clear ability to identify drought trends linked to anthropogenic forcing^{33,34}. While some studies have explored drought projections using SPEI^{35,36}, only a few have examined global-scale trends, indicating an increase in drought severity associated with global warming³⁷. Other global studies have assessed drought trends using SPEI with observational data but did not evaluate the influence of AED on drought severity or address uncertainties in precipitation and AED datasets—critical limitations for drawing robust conclusions^{19,38,39}. Only one study² has examined the role of anthropogenic climate change on drought severity using CMIP6 simulations, but it introduces significant uncertainties due to the limitations of model-based approaches. While SPEI has been widely used to assess drought trends, this study is the first to quantify, at a global scale and based on observations, the role of increasing AED in drought severity.

Additionally, it evaluates uncertainties in global datasets, offering a more comprehensive perspective on this critical issue.”

Detailed review.

Abstract

I suggest moving the statements regarding the uncertainties in previous global drought assessments to the introduction.

Authors' response:

- Thank you for your suggestion. We have moved the statement to the introduction:

Lines 107-110: “Nevertheless, previous studies had highlighted significant uncertainties in global-scale drought assessments and in the determination of the role of AED on drought severity, largely due to the choice of models for AED and meteorological forcing dataset^{3,4,21,22}.”

Introduction

The presentation of the topic is well written, though the authors may want to expand the debate over AED, not only citing the papers (as they properly do), but also adding more reasoning and maybe comparisons between different opinions/papers on AED.

Authors' response:

- We agree and are aware of the existing debate on the role of AED effects on drought severity, which have produced that there are different views and even perspectives on drought trends, mostly motivated by the differences in drought projections between metrics directly obtained from earth system models (e.g., streamflow, soil moisture, leaf area index) and drought indices (e.g., the SPEI or the PDSI) that are based on different meteorological variables that in addition to precipitation are used to calculate AED. To address this, we have revised the introduction and included additional discussion in Lines 93–104 of the manuscript.

“Whilst drought can be characterized in many ways to reflect different meteorological, hydrological and ecological drivers, consideration of the influence of AED with respect to precipitation is crucial to understand how climate change is impacting on changes in drought. Some studies suggest that AED-based drought metrics may overestimate severity compared to hydrological and ecological indicators(Berg and Sheffield 2018). However, this mainly stems from uncertainties in Earth system model projections and the physiological effects of atmospheric CO₂ on evaporation(Vicente-Serrano et al. 2022; Cook et al. 2020). Methodological challenges also affect comparisons between drought metrics, but applying consistent statistical approaches shows stronger agreement between AED-inclusive indices (Garrido-Perez et al. 2024b). Increasing evidence highlights AED's role in amplifying ecological drought severity via evaporation(Bevacqua et al. 2024b). Given AED's recent rise and projected increase due to anthropogenic warming(Vicente-Serrano et al. 2022; Douville and Willett 2023), assessing its contribution to drought severity is essential for adaptation planning.”

If you choose those two datasets for precipitation, I would expect more robust motivation than just a few papers, also some statistics.

Authors' response:

- MSWEP and CHIRPS have been comprehensively evaluated and shown to be robust, and generally out-perform other global datasets, though are not universally accurate (Beck et al. 2017, Gebrechorkos et al. 2023). They also provide differences in approaches (e.g. use of data from different types of satellite sensors, and from differing regional station databases), which provides some independence in the datasets. We have previously assessed these two datasets using statistical measures like Kling–Gupta Efficiency (KGE), Nash-Sutcliffe Efficiency (NSE), bias, and error metrics (Beck et al. 2017, Gebrechorkos et al. 2023).
- Based on your recommendation, we have added a paragraph in the "Global Climate and AED Datasets" section (Lines 489–521) elaborating on the similarities and differences between MSWEP and CHIRPS, along with our rationale for selecting both. Additionally, we have included correlation and average monthly difference maps in Extended Data Fig. 9.

“CHIRPS and MSWEP were chosen as they generally out-perform other similar gridded precipitation datasets when compared to ground observations (Gebrechorkos et al. 2024; Beck et al. 2017). CHIRPS (0.05°) is particularly designed for monitoring droughts and detecting environmental changes, providing daily precipitation estimates from 1981 to present. It combines satellite-derived Climate Hazards Center Infrared Precipitation (CHIRP) and the Climate Hazards Group Precipitation Climatology (CHPclim) with ground station data from the Global Historical Climate Network (GHCN) and many other sources. The CHIRPS product benefits from a high degree of homogeneity, provided by its simple but consistent foundation of geostationary thermal infrared satellite observations. CHIRPS also incorporates unique observation inputs from Africa, Latin America and Central America. MSWEP (0.1°) has been designed with both accuracy and homogeneity in mind, providing 3-hourly precipitation estimates from 1979 to present. It integrates daily observations from over 77,000 stations from various national and international data sources, satellite estimates from infrared- and microwave-based satellite datasets, and reanalysis data, offering accurate global precipitation data from 1979 to present. Both CHIRPS and MSWEP have previously been evaluated globally using statistical metrics such as Kling–Gupta Efficiency (KGE) and Nash-Sutcliffe Efficiency (NSE), as well as various bias and error metrics (Gebrechorkos et al. 2024; Beck et al. 2017). For instance, MSWEP outperformed 22 other global precipitation datasets in capturing daily precipitation from 76,086 gauging stations and in driving hydrological models across 9,053 catchments (Beck et al. 2017). Additionally, both MSWEP and CHIRPS were found to outperform other high-resolution gauge-based datasets in modelling daily, monthly, and annual streamflow across 1,825 streamflow gauges (Gebrechorkos et al. 2024). However, both datasets remain subject to inherent uncertainties, and therefore, considering both helps reduce biases and obtain more reliable estimates, given that they are somewhat independent. For example, they differ in their data sources with CHIRPS using only geostationary thermal infrared observations, whilst MSWEP also uses microwave observations, and they use different sets of

station data to correct locally. Despite these differences, the monthly correlation between MSWEP and CHIRPS shows a high correlation across most regions, except for Central Asia (Extended Data Fig. 9a). The average monthly difference between the two datasets varies spatially, reaching up to ± 40 mm (Extended Data Fig. 9c). Notably, larger discrepancies occur in regions such as the Amazon, Central Africa, and parts of Southeast Asia. Such convergence between the two products helps reduce concerns about the uncertainties due to different approaches and changes in the constellation of Earth-observing satellites that can affect the robustness of their representation of changes over time.”

Extended Data Fig. 9: Monthly correlation between CHIRPS and MSWEP for precipitation, and GLEAM and hPET for AED, along with their differences during 1981–2022. The precipitation correlation is limited to latitudes up to 50°N due to the availability of CHIRPS data only up to this latitude. Panel a) shows the correlation between MSWEP and CHIRPS, while panel b) displays the correlation between GLEAM and hPET AED. Panels c) and d) show the average monthly difference between MSWEP and CHIRPS, and GLEAM and hPET, respectively.

It is not immediately clear - without reading the reference - which are the four datasets part of the HRSPEI data. Moreover, it is not clear how you are going to weight or divide the single datasets to combine them in overall results. I am sure that such details will come later in the paper, but a simple line with introductory methodology on this aspect could help.

Authors' response:

- Thank you for the suggestion. The four datasets were developed from the combinations of precipitation from CHIRPS or MSWEP and AED from the Global Land Evaporation Amsterdam Model (GLEAM) or the hourly Potential Evapotranspiration (hPET) dataset, i.e. two datasets based on CHIRPS with AED from GLEAM or hPET, and two based on MSWEP with AED from GLEAM or hPET. As recommended, we have added this in the Introduction, lines 159–162.

“We developed four global, high-resolution (0.05°) SPEI datasets for 1981–2022 using precipitation from CHIRPS or MSWEP, combined with AED from Global Land Evaporation Amsterdam Model (GLEAM) or hourly Potential Evapotranspiration (hPET).”

- The HRSPEI is calculated as the mean of these four datasets. For latitudes up to 50°N, all four datasets are used, but for latitudes above 50°N, only the MSWEP-based indices are included, as CHIRPS is limited to 50°S–50°N. We have added a clarifying line in the Drought Index and data sections of the methodology. To develop the HRSPEI, we use all datasets equally, without weighting, to calculate the mean. Both the mean (HRSPEI) and results from individual datasets are presented for clarity. In the “Drought Index” section (lines 448–456), the following text is added:

“The resulting four indices: MSWEP_GLEAM, MSWEP_hPET, CHIRPS_GLEAM, and CHIRPS_hPET, were developed at a spatial resolution of 0.05° for the period 1981–2022. The 0.1° resolution datasets were first interpolated to match the resolution of CHIRPS using bilinear interpolation. Additionally, we developed an ensemble mean (HRSPEI) based on all four datasets. For latitudes above 50°N, the mean is derived from MSWEP_GLEAM and MSWEP_hPET, as CHIRPS data is only available up to ±50° latitude. AED and AED variability in high-latitude areas > 50°N is generally small, and changes in AED, even at high percentages, result in low absolute magnitudes, making SPEI less sensitive to AED in these regions. ”

Trends in.

What is the baseline period used to compute SPEI for the various datasets? Are they homogeneous on this aspect? This is a crucial aspect.

Authors’ response:

- We agree that the baseline period is a crucial aspect, as differences in data coverage can lead to discrepancies in the computed indices, one of the main sources of uncertainty in previous studies as explained in the paper. To ensure consistency across the datasets, we used the period from 1981 to 2022 for the computation of all four SPEI datasets. This approach ensures homogeneity in terms of the baseline period across all datasets. As explained in the paper, varying calibration periods introduce additional uncertainties. To clarify this point, we have added a line in the "Drought Index" section (Lines 437–441).

“The SPEI indices were calculated using the entire 1981–2022 period as a baseline, ensuring that the full range of variability in the input data is captured. Unlike other drought indices, SPEI does not require a predefined baseline or calibration period, as it standardizes the data directly from the input time series, ensuring consistency across datasets and time scales.”

Why cutting lands over 50°N when averaging trends? Is this due to lack of reliable data at high Latitudes? Or because some datasets do not have data there (which is a relevant limitation)? You should motivate such decisions in the core text, not only in the figure captions.

Authors’ response:

- Thank you for pointing this out. As explained above, the primary reason for limiting the time series analysis to latitudes up to 50°N is the restricted spatial coverage of CHIRPS, which only extends to 50°N. To ensure consistency, we averaged trends using all four datasets only within this latitude range. However, for spatial trend analysis, we utilized MSWEP-based indices to extend coverage beyond 50°N. Additionally, AED variability in high-latitude regions is relatively small. Even substantial changes in AED result in low absolute magnitudes, making SPEI less sensitive in these areas. To clarify this, we have added a few lines with appropriate references in the "Drought Index" section (Lines 448–456) and revised the figure legend for improved transparency.

“The resulting four indices: MSWEP_GLEAM, MSWEP_hPET, CHIRPS_GLEAM, and CHIRPS_hPET, were developed at a spatial resolution of 0.05° for the period 1981–2022. The 0.1° resolution datasets were first interpolated to match the resolution of CHIRPS using bilinear interpolation. Additionally, we developed an ensemble mean (HRSPEI) based on all four datasets. For latitudes above 50°N, the mean is derived from MSWEP_GLEAM and MSWEP_hPET, as CHIRPS data is only available up to ±50° latitude. AED and AED variability in high-latitude areas > 50°N is generally small, and changes in AED, even at high percentages, result in low absolute magnitudes, making SPEI less sensitive to AED in these regions.”

I would suggest including the data on trends/year in a table for a better reading.

Authors' response:

- We have added a summary table for SPEI trends and areas based on HRSPEI, CRU-TS, and ERA5 for 1981–2022, as well as for 1950–1980 (ERA5 and CRU-TS), in Extended Data Fig. 1D as shown below.

Extended Data Fig. 1: Trends in 6 month SPEI for the period 1981–2022. Panels a), b), c), and d) display the 6-month SPEI trends (z-units year⁻¹) derived from the MSWEP_hPET, MSWEP_GLEAM, CHIRPS_hPET, and CHIRPS_GLEAM datasets, respectively. Non-significant trends (P-value > 0.05) are set to zero to improve visualization clarity. The analysis excludes dryland regions with an average annual rainfall of less than 180 mm. Panel e) shows the quasi-global (50°S to 50°N) average SPEI, while panel f) summarizes the SPEI trends and areas under drought conditions based on the HRSPEI, CRU-TS, and ERA5 datasets.

You said that "in 2022, annual precipitation across Europe dropped by up to 35% below average". Averaged over which period?

Authors' response:

- Thank you for pointing this out. The period used for comparison is the study period of this work, 1981–2022. We have updated the text, lines 226–228, to specify this as:

“In 2022, annual precipitation across Europe dropped by up to 35% below the 1981–2022 average, while AED rose by up to 40% (Extended Data Fig. 5).”

2022 was an exceptional year for drought, as you clearly mention it in the title. I would like to see a more detailed discussion on its drought characteristics whenever you mention it, not in the extended data, but directly in the core text.

Authors' response:

- We agree with this point. As recommended by both reviewers, we have removed 2022 from the title as the focus of the paper is more on the long-term trends. Nevertheless, 2022 shows the large role of AED on drought severity and associated hydrological and ecological impacts. In the revised discussion section we have incorporated further discussion of the 2022 drought, highlighting its exceptional characteristics and its underlying drivers. For example, the following is added in lines 391-400:

“In Europe, the severity of the 2022 drought event can be largely attributed to anthropogenic warming, since the anomalies observed in streamflow and soil moisture cannot be explained by the precipitation deficit alone, but mostly by enhanced AED, which increased water losses by evaporation (Bevacqua et al. 2024b; Garrido-Perez et al. 2024b). Moreover, ecological drought severity recorded in Europe's natural forests cannot be fully explained without considering the influence of high temperatures and AED on plant physiology. In the absence of formal attribution studies in other regions of the world that experienced drought in 2022, the attribution in Europe and the increase in severity globally driven by enhanced AED as shown in this study, suggests that it is reasonable to conclude that anthropogenic global warming likely contributed to exacerbate global drought severity in 2022.”

Drivers of..

Why not comparing SPI and SPEI, other than using AED_{clm} and AED?

Authors' response:

- This was our first attempt, as we understand this to be the most logical approach. However, it introduces methodological challenges because the recommended probability distributions used to calculate both indices are different. SPI is calculated using a Gamma distribution, which is bounded in the lower tail at zero, whereas SPEI is calculated using a log-logistic distribution. Even if the data perfectly fits the selected distribution, differences can arise, particularly in the lower tail, which is critical for defining the severity of droughts (Vicente-Serrano and Beguería 2016). Moreover, SPI calculations show significant uncertainties in water-limited regions characterized by low precipitation (Wu et al. 2007), and the high frequency of zero values in these regions is also a challenge (Stagge et al. 2015). For these reasons, isolating the particular role of AED trends on drought severity would introduce uncertainties if we were to compare SPI and SPEI.

On the contrary, calculating a new version of the SPEI while maintaining AED as constant solves these problems, as the same probability distribution is used for both versions. This approach also resolves the issue of assessing drought severity in water-

limited regions with low precipitation by providing continuous values (potentially between $-\infty$ and $+\infty$) in the difference between P and AED. Even in the most hyperarid regions of the world, where total precipitation tends toward zero, the drought index that results from the difference between P and AED would resemble the approach used to calculate the Evaporative Demand Drought Index (EDDI) (Hobbins et al. 2016), which is based solely on AED and for which the log-logistic distribution has also shown better performance in calculations (Noguera et al. 2022). Recently, (Garrido-Perez et al. 2024a) used this approach to assess the severity of the 2022 drought in Europe, demonstrating its capacity to evaluate the effect of warming on drought severity. In fact, the use of this approach in our study is a clear methodological improvement for quantifying the role of AED on drought severity, and we are confident it will be used in future studies.

Through the text, it is not fully clear which AED method applies to the various SPEI deriving from different datasets.

Authors' response:

- Thank you for your observation. As explained above, the updated GLEAM dataset now utilizes the original Penman equation, which is similar to the FAO-56 Penman-Monteith equation used in developing hPET. In the methodology section, under "Atmospheric Evaporative Demand (AED)," we have provided a detailed explanation of the AED methods applied to each dataset. Specifically, GLEAM employs the Penman equation (Eq. 1), while hPET is based on the FAO-56 Penman-Monteith equation (Eq. 1).
- We have revised the methodology section (Lines 526–537) to clarify these distinctions for improved transparency:

“In addition, the AED from the Global Land Evaporation Amsterdam Model (GLEAM, version 4.2a) is derived using the Penman's original equation (Eq. 2), using satellite and reanalysis datasets⁷⁹. GLEAM v4.2a is available at a 0.1° spatial resolution and covers the period 1980–2022. hPET is based on a FAO Penman–Monteith equation which computes reference crop evaporation by assuming certain surface and aerodynamic characteristics that are constant in time. In contrast, GLEAM4 calculates aerodynamic conductance as a dynamic variable depending on ecosystem characteristics and local meteorology and therefore is space and time dependent. Nonetheless, given the dominant influence of radiative forcing and atmospheric aridity in both computations, their estimates are overall similar. The monthly average difference between GLEAM and hPET is up to ± 3 mm (Extended Data Fig. 9d), and their correlation exceeds 0.9 across 91% of the global land surface (Extended Data Fig. 9b).

The discussion on Figure 4 is too short, I feel this part has relevant potential for the readers.

Authors' response:

- Thank you for acknowledging the importance of Figure 4. We have added more discussion to the manuscript (lines 315–324) as recommended.

“Overall, even though precipitation accounts for 60% of the global average SPEI trend during 1981–2022, the role of AED, contributing 40%, is substantial (Fig. 4). This is especially notable considering the stronger sensitivity of SPEI to precipitation than to AED in most land regions (Tomas-Burguera et al. 2020). Furthermore, although the trend contribution covers 1981–2022, the impact of AED has become increasingly evident since 2000 (Fig. 3a). In

Africa, Australia, and the drylands of North and South America, the influence of AED is particularly pronounced, contributing up to 65% to drought trends during 1981–2022. Specifically, AED accounts for 44% of the drought trend in Africa and 51% in Australia, playing a significant role in intensifying drought severity in these regions. In contrast, the contribution of AED to drought trends in North and South America, Europe, and Asia is around 30%”.

Acceleration of..

Also here, as in the most parts of the core paper, the key results could be grouped in a Table, this may improve the reading.

Authors’ response:

- Thank you for your suggestion to present the results in a table format. We agree that this would enhance readability, and we have included a table summarizing the key trends in Extended Data Fig. 8, as we are limited to four figures in the main text.
- The summary comparing areas impacted by drought based on AED versus AEDclm for the periods 1981–2017 and 2018–2022 is provided in Extended Data Fig. 8h, as shown below:

Region	Area in drought (%) based on AED		Area in drought (%) based on AEDclm	
	1981-2017	2018-2022	1981-2017	2018-2022
Australia	13.2	28.8	14	19
Southern South America	10.5	27.5	10.9	17.6
Western USA	11.6	28	12	19
East Africa	12	21	12.5	11.3
Northern Asia	15.3	27.6	16.5	16.6
Europe	26	41	27.7	26
Global	15.5	27	15.7	17

Extended Data Fig. 8: Annual average areas affected by drought by region and globally for 1981–2022. The time series shows the annual percentage of areas in drought (SPEI < -1) for a) Australia, b) East Africa, c) Europe, d) Northern Asia, e) Southern South America, f) Western USA, and g) globally, based on the 6-month SPEI. The red lines represent the 6-month HRSPEI calculated using observed AED and precipitation, while the blue lines indicate the HRSPEI derived from observed precipitation and climatological AED (AEDclm). The dashed vertical blue lines indicate the period from 2018 to 2022, indicating the rise in areas in drought. figure h) provides the summary of the areas affected by drought in 2018-2022 compared to 1981-2017 based on observed AED and AEDclm.

Again, the record-breaking 2022 is even included in the title. I want to read more about it, at least I would like to see a dedicated figure-

Authors' response:

- As discussed above, to enhance the clarity and focus of the paper, we have removed the word "record-breaking 2022" from the title. However, we have added a global time series of areas in drought based on observed AED and climatological AED (AED_{clm}) for the period 1981–2022 in Extended Data Fig. 9, which clearly highlights the intensification of drought areas in recent years, particularly the 2022 drought globally and in certain regions, such as Europe and East Africa. Additionally, Figure 1d illustrates the global 6-month SPEI for 2022, highlighting the intensity of the drought in key regions, including Europe, East Africa, Southern South America, Western USA, and large parts of Asia. Furthermore, Extended Data Fig. 4 provides a time series of the percentage of areas impacted by extreme and severe droughts, revealing a heightened impact during the period 2018–2022 compared to 1981–2017. We have also included a discussion about the 2022 drought, as explained above.
- We hope that the additional figures and information will provide sufficient evidence of the increase in drought severity and the role of AED, with the year 2022 as an example.

Conclusions

They are a summary of your findings; I feel that you could expand it, also, including why this study could be of a higher relevance compared to previous global drought assessments.

Authors' response:

- We appreciate your feedback and have revised and expanded the discussion section to highlight the relevance and novelty of our study in lines 375–417 as:

“Discussion

According to the Standardized Precipitation Evapotranspiration Index (SPEI), over the past 42 years (1981–2022), global drought severity has intensified. In the last 5–10 years, this trend has accelerated as a consequence of the strong increase in AED, which is directly related to global warming and increased vapour pressure deficit (Douville and Willett 2023), as the water supply to the atmosphere is not enough to compensate for the large temperature increase. Some recent studies had also suggested an increase in the severity of drought events over large land areas based on metrics like modelled soil moisture (Padrón et al. 2020) and the Palmer Drought Severity Index (Dai and Zhao 2017; Song et al. 2020), all of them sensitive to changes in the AED. Nevertheless, in our study, we have quantified the contribution of AED to worsening drought conditions, which has been up to 60% in some regions, particularly in Africa, Australia, Western USA, and Southern South America. Moreover, changes in AED have exacerbated the drying trend globally, particularly in the last decade. The year 2022 specifically was a record-breaking year for drought severity and extent in Europe and East Africa. In Europe, the severity of the 2022 drought event can be largely attributed to anthropogenic warming, since the anomalies observed in streamflow and soil moisture cannot be explained

by the precipitation deficit alone, but mostly by enhanced AED, which increased water losses by evaporation (Bevacqua et al. 2024b; Garrido-Perez et al. 2024b). Moreover, ecological drought severity recorded in Europe's natural forests cannot be fully explained without considering the influence of high temperatures and AED on plant physiology. In the absence of formal attribution studies in other regions of the world that experienced drought in 2022, the attribution in Europe and the increase in severity globally driven by enhanced AED as shown in this study, suggests that it is reasonable to conclude that anthropogenic global warming likely contributed to exacerbate global drought severity in 2022.

In comparison to previous studies analysing recent drought trends based on atmospheric drought indices that use AED in calculations (Chiang et al. 2021; Dai and Zhao 2017; Song et al. 2020; Vicente-Serrano et al. 2022), this study has isolated for the first time the effect of AED on drought severity and in addition our study has also reduced uncertainties given the use of high spatial resolution and multi-source data, which allows for a clearer understanding of drought intensification. The observed increase in drought severity aligns with associated impacts on agricultural, environmental, and hydrological systems, as seen in events like the 2022 European drought, which contributed to enhanced tree mortality (Forzieri et al. 2022), increased forest fires (Alizadeh et al. 2021), and long-term soil moisture decline (Padrón et al. 2020). Although the SPEI is an atmospheric drought index that effectively captures the effects of precipitation and AED on drought severity, it may represent drought-related impacts very effectively (Bachmair et al. 2018; Vicente-Serrano et al. 2012). However, further studies are needed, considering variables such as soil moisture, vegetation stress, and hydrological flows for better understanding of the broader impacts of the observed changes on ecosystems and human activities (Vicente-Serrano et al. 2022). Moreover, the observed acceleration of drought trends in the past few years aligns with future climate projections that indicate further increases in drought severity due to projected warming (Spinoni et al. 2020; Naumann et al. 2018), which warns of the need for better socioeconomic and environmental adaptation measures to reduce drought impacts and improve global drought. ”

Methods

Which baseline did you use to compute the SPEI over different periods?

Authors' response:

- As mentioned above, the four SPEI datasets were computed for the period 1981–2022, and the SPEI indices were derived using the entire period for the calculation. This approach ensures that the indices account for the full range of variability observed across the time series, capturing the long-term trends and the most recent data for a comprehensive analysis of drought conditions. To make this part clearer, we have added the following in the Methods, Drought index section: lines: 437-441:

“The SPEI indices were calculated using the entire 1981–2022 period as a baseline, ensuring that the full range of variability in the input data is captured. Unlike other drought indices, SPEI does not require a predefined baseline or calibration period, as it standardizes the data directly from the input time series, ensuring consistency across datasets and time scales.”

The choice of thresholds for categories of SPEI is a bit arbitrary, could you motivate why using exactly those Numbers instead of the classic -1, -2, etc.?

Authors' response:

- We understand the concern regarding the choice of thresholds. The thresholds proposed by (McKee et al. 1993) for drought classification are inherently arbitrary. For instance, a z-value of -1, which marks the boundary between moderately and severely dry conditions, corresponds to a cumulative probability of 0.158 and a return period of 6.301 years under a normal distribution. Similarly, a z-value of -2, delineating the boundary between severely and extremely dry conditions, corresponds to a cumulative probability of 0.0228 and a return period of 43.859 years.
- We believe that these thresholds may not accurately reflect the expected frequency of drought events. Specifically, the threshold for moderately dry conditions results in an excessively high frequency of occurrence, given its low return period. Conversely, the threshold for extremely dry conditions implies an exceptionally low frequency of occurrence, owing to its very high return period.
- To address this discrepancy, we opted to focus on return periods rather than z-values, as this approach aligns more closely with statistical coherence, as suggested by (Agnew 2000). Based on this reasoning, we selected a threshold of -1.43 to define the boundary between moderately and severely dry conditions, as it corresponds to a return period of about 13 years. This threshold better represents the expected frequency of severe droughts. Additionally, we established a threshold of -1.82, corresponding to a return period of about 28 years, to define the boundary between severely and extremely dry conditions. In the context of global warming, extreme drought events are occurring with increasing frequency. While these thresholds remain arbitrary to some extent—since any threshold inherently involves some degree of subjectivity—they are more consistent with observed and expected drought frequencies in the current climatic scenario. This refinement helps ensure that the classification of drought conditions aligns more closely with both statistical principles and real-world observations. While these thresholds remain somewhat subjective, they are more consistent with real-world observations and statistical principles. Additionally, they are commonly used in the literature (e.g., Danandeh Mehr and Vaheddoost 2020; Worku 2024; Prabnakorn et al. 2018; Danandeh Mehr et al. 2020; WMO 2012; Hayes 2006; McKee et al. 1993; Nam et al. 2015; Yu et al. 2023; Liu et al. 2021).

If you use different methods to compute PET, one might questions if this affects the results. Why not using both methods for all the SPEI computed? Is this possible or one or more datasets lack input data to do that?

Authors' response:

- We appreciate the reviewer's question regarding the use of different methods for computing Potential Evapotranspiration (PET). Indeed, the choice of PET calculation method can influence the results, and we acknowledge this potential impact. For the four high-resolution SPEI datasets, we in fact used both methods to compute AED. As stated above, we have updated the AED from GLEAM to Version 4.2a, which is based on the Penman method, replacing the previous version computed with the Priestley-Taylor equation.

- Similarly, the hPET dataset was developed using the FAO-56 Penman-Monteith equation, which shares a strong theoretical foundation with the Penman equation employed in the latest version of GLEAM (v4.2a). To ensure consistency and address potential uncertainties, we computed SPEI using both AED datasets—GLEAM (Version 4.2a) and hPET—with two precipitation datasets (MSWEP and CHIRPS). This approach resulted in four combinations: MSWEP-GLEAM, MSWEP-hPET, CHIRPS-GLEAM, and CHIRPS-hPET. For the ERA5 and CRU-TS datasets, we used the FAO-56 Penman-Monteith equation exclusively. The Penman equation considers a wider range of climate variables, making it a more robust choice for accurately estimating AED, particularly in regions with complex climate patterns. While the GLEAM model's AED is widely used and has been validated with observations, we chose to use both datasets—GLEAM and hPET—because they offer complementary strengths. The use of both methods allows us to account for the uncertainty and variability associated with different input datasets and approaches for computing AED. We hope this clarifies. As explained above the correlation between the hPET and GLEAM is greater than 0.9 across 91% of the global land, with an average difference of 3mm (Extended Data Fig. 9). Also, both AED and GLEAM show a similar trend in large part of the world (Extended Data Fig. 7)

Figures and Tables

Captions are very long, consider reducing.

Authors' response:

- Thank you for your suggestion. We understand that the captions are quite long, and we have used this space to provide additional context due to the word limit in the main text. However, in response to your feedback, we have made efforts to reduce the length of the caption in Figure 3 for clarity and conciseness, while still retaining essential details.

Figure 3 - I would show non-significant trends, maybe with dashed lines, instead of setting them to 0.

Authors' response:

- Thank you for the recommendation. We attempted to represent the non-significant trends using dashed lines, as suggested. However, due to the high-resolution nature of our data, this approach affected the visibility and quality of the maps. While we agree with your suggestion in principle, we believe setting non-significant trends to 0 is a more effective method for maintaining clarity, particularly with high-resolution datasets. We recognize that dashed lines can work well for coarser datasets, but in this case, it may compromise the readability of the results.

Figure 4 is the one of the most interesting parts of the paper, but panels a) and b) are useless, if they sum to 100%, just use a single colour scale focusing on one of them. Same applies to the histograms.

Authors' response:

- Thank you for your feedback and for highlighting the importance of this part of the work. We agree that the finding is crucial for enhancing the understanding of the impact of AED on droughts. While it is true that panels a) and b) sum to 100%, we included both for ease of comparison between the contributions of precipitation and AED. This approach allows readers to directly compare their spatial patterns.
- In response to your suggestion, and the suggestion by referee #1, we have revised the figure by using a single continuous colour scale, as opposed to a diverging colour scale, to improve clarity and focus.

Fig. 4: Percentage of contribution (%) of AED and precipitation (Pr) to 6-month SPEI trends. Figures a) show the percentage of contribution of AED and b) show the percentage of contribution of Pr for the observed changes in 6-month HRSPEI during 1981–2022. The

contributions are computed by calculating the difference between the observed trend and the trend based on the climatological values of AED (AEDclm) and precipitation (Prclm). The contribution of AED is determined by the difference between the trend using observed AED and Pr and the trend using observed Pr and AEDclm. Similarly, the contribution of Pr is calculated as the difference between the trend using observed Pr and AED and the trend using observed AED and Prclm. The percentage contribution of each factor is then calculated as the absolute value of the difference divided by the total absolute difference, providing a relative measure of each factor's influence on the observed trend. The lower panel (c) provides the regional and global average contribution of Pr and AED to the changes in 6-month SPEI.

References:

- Agnew, C. T., 2000: Using the SPI to Identify Drought. *Drought Network News (1994-2001)*,.
- Alizadeh, M. R., J. T. Abatzoglou, C. H. Luce, J. F. Adamowski, A. Farid, and M. Sadegh, 2021: Warming enabled upslope advance in western US forest fires. *Proceedings of the National Academy of Sciences*, **118**, e2009717118, <https://doi.org/10.1073/pnas.2009717118>.
- Bachmair, S., M. Tanguy, J. Hannaford, and K. Stahl, 2018: How well do meteorological indicators represent agricultural and forest drought across Europe? *Environ. Res. Lett.*, **13**, 034042, <https://doi.org/10.1088/1748-9326/aaafda>.
- Beck, H. E., and Coauthors, 2017: Global-scale evaluation of 22 precipitation datasets using gauge observations and hydrological modeling. *Hydrology and Earth System Sciences*, **21**, 6201–6217, <https://doi.org/10.5194/hess-21-6201-2017>.
- Berg, A., and J. Sheffield, 2018: Climate Change and Drought: the Soil Moisture Perspective. *Curr Clim Change Rep*, **4**, 180–191, <https://doi.org/10.1007/s40641-018-0095-0>.
- Bevacqua, E., O. Rakovec, D. L. Schumacher, R. Kumar, S. Thober, L. Samaniego, S. I. Seneviratne, and J. Zscheischler, 2024a: Direct and lagged climate change effects intensified the 2022 European drought. *Nature Geoscience*, **17**, 1100–1107, <https://doi.org/10.1038/s41561-024-01559-2>.
- , ———, ———, ———, ———, ———, ———, and ———, 2024b: Direct and lagged climate change effects intensified the 2022 European drought. *Nat. Geosci.*, **17**, 1100–1107, <https://doi.org/10.1038/s41561-024-01559-2>.
- Chiang, F., O. Mazdiyasn, and A. AghaKouchak, 2021: Evidence of anthropogenic impacts on global drought frequency, duration, and intensity. *Nat Commun*, **12**, 2754, <https://doi.org/10.1038/s41467-021-22314-w>.
- Cook, B. I., J. S. Mankin, K. Marvel, A. P. Williams, J. E. Smerdon, and K. J. Anchukaitis, 2020: Twenty-First Century Drought Projections in the CMIP6 Forcing Scenarios. *Earth's Future*, **8**, e2019EF001461, <https://doi.org/10.1029/2019EF001461>.
- Dai, A., and T. Zhao, 2017: Uncertainties in historical changes and future projections of drought. Part I: estimates of historical drought changes. *Climatic Change*, **144**, 519–533, <https://doi.org/10.1007/s10584-016-1705-2>.

- Danandeh Mehr, A., and B. Vaheddoost, 2020: Identification of the trends associated with the SPI and SPEI indices across Ankara, Turkey. *Theor Appl Climatol*, **139**, 1531–1542, <https://doi.org/10.1007/s00704-019-03071-9>.
- , A. U. Sorman, E. Kahya, and M. Hesami Afshar, 2020: Climate change impacts on meteorological drought using SPI and SPEI: case study of Ankara, Turkey. *Hydrological Sciences Journal*, **65**, 254–268, <https://doi.org/10.1080/02626667.2019.1691218>.
- Douville, H., and K. M. Willett, 2023: A drier than expected future, supported by near-surface relative humidity observations. *Science Advances*, **9**, eade6253, <https://doi.org/10.1126/sciadv.ade6253>.
- Dracup, J. A., K. S. Lee, and E. G. Paulson Jr., 1980: On the statistical characteristics of drought events. *Water Resources Research*, **16**, 289–296, <https://doi.org/10.1029/WR016i002p00289>.
- Fleig, A. K., L. M. Tallaksen, H. Hisdal, and S. Demuth, 2006: A global evaluation of streamflow drought characteristics. *Hydrology and Earth System Sciences*, **10**, 535–552, <https://doi.org/10.5194/hess-10-535-2006>.
- Forzieri, G., V. Dakos, N. G. McDowell, A. Ramdane, and A. Cescatti, 2022: Emerging signals of declining forest resilience under climate change. *Nature*, **608**, 534–539, <https://doi.org/10.1038/s41586-022-04959-9>.
- Garrido-Perez, J. M., S. M. Vicente-Serrano, D. Barriopedro, R. García-Herrera, R. Trigo, and S. Beguería, 2024a: Examining the outstanding Euro-Mediterranean drought of 2021–2022 and its historical context. *Journal of Hydrology*, **630**, 130653, <https://doi.org/10.1016/j.jhydrol.2024.130653>.
- , ———, ———, ———, ———, and ———, 2024b: Examining the outstanding Euro-Mediterranean drought of 2021–2022 and its historical context. *Journal of Hydrology*, **630**, 130653, <https://doi.org/10.1016/j.jhydrol.2024.130653>.
- Gebrechorkos, S. H., and Coauthors, 2024: Global-scale evaluation of precipitation datasets for hydrological modelling. *Hydrology and Earth System Sciences*, **28**, 3099–3118, <https://doi.org/10.5194/hess-28-3099-2024>.
- Hayes, M. J., 2006: Drought Indices. *Van Nostrand's Scientific Encyclopedia*, John Wiley & Sons, Ltd.
- Hobbins, M. T., A. Wood, D. J. McEvoy, J. L. Huntington, C. Morton, M. Anderson, and C. Hain, 2016: The evaporative demand drought index. Part I: Linking drought evolution to variations in evaporative demand. *Journal of Hydrometeorology*, **17**, 1745–1761, <https://doi.org/10.1175/JHM-D-15-0121.1>.
- Lenssen, N. J. L., G. A. Schmidt, J. E. Hansen, M. J. Menne, A. Persin, R. Ruedy, and D. Zyss, 2019: Improvements in the GISTEMP Uncertainty Model. *Journal of Geophysical Research: Atmospheres*, **124**, 6307–6326, <https://doi.org/10.1029/2018JD029522>.

- Liu, C., C. Yang, Q. Yang, and J. Wang, 2021: Spatiotemporal drought analysis by the standardized precipitation index (SPI) and standardized precipitation evapotranspiration index (SPEI) in Sichuan Province, China. *Sci Rep*, **11**, 1280, <https://doi.org/10.1038/s41598-020-80527-3>.
- Lorenzo-Lacruz, J., E. Moñan-Tejeda, S. M. Vicente-Serrano, and J. I. López-Moreno, 2013: Streamflow droughts in the Iberian Peninsula between 1945 and 2005: Spatial and temporal patterns. *Hydrology and Earth System Sciences*, **17**, 119–134, <https://doi.org/10.5194/hess-17-119-2013>.
- McKee, T. B., N. J. Doesken, and J. Kliest, 1993: The relationship of drought frequency and duration to time scales. In Proceedings of the 8th Conference of Applied Climatology, 17-22 January, Anaheim, CA. American Meteorological Society, Boston, MA. 179–184
http://www.droughtmanagement.info/literature/AMS_Relationship_Drought_Frequency_Duration_Time_Scales_1993.pdf (Accessed February 10, 2018).
- Nam, W.-H., M. J. Hayes, M. D. Svoboda, T. Tadesse, and D. A. Wilhite, 2015: Drought hazard assessment in the context of climate change for South Korea. *Agricultural Water Management*, **160**, 106–117, <https://doi.org/10.1016/j.agwat.2015.06.029>.
- Naumann, G., and Coauthors, 2018: Global Changes in Drought Conditions Under Different Levels of Warming. *Geophysical Research Letters*, **45**, 3285–3296, <https://doi.org/10.1002/2017GL076521>.
- Noguera, I., S. M. Vicente-Serrano, F. Domínguez-Castro, and F. Reig, 2022: Assessment of parametric approaches to calculate the Evaporative Demand Drought Index. *International Journal of Climatology*, **42**, 834–849, <https://doi.org/10.1002/joc.7275>.
- Padrón, R. S., and Coauthors, 2020: Observed changes in dry-season water availability attributed to human-induced climate change. *Nat. Geosci.*, **13**, 477–481, <https://doi.org/10.1038/s41561-020-0594-1>.
- Prabnakorn, S., S. Maskey, F. X. Suryadi, and C. de Fraiture, 2018: Rice yield in response to climate trends and drought index in the Mun River Basin, Thailand. *Science of The Total Environment*, **621**, 108–119, <https://doi.org/10.1016/j.scitotenv.2017.11.136>.
- Song, X., Y. Song, and Y. Chen, 2020: Secular trend of global drought since 1950. *Environ. Res. Lett.*, **15**, 094073, <https://doi.org/10.1088/1748-9326/aba20d>.
- Spinoni, J., and Coauthors, 2020: Future Global Meteorological Drought Hot Spots: A Study Based on CORDEX Data. *Journal of Climate*, **33**, 3635–3661, <https://doi.org/10.1175/JCLI-D-19-0084.1>.
- Stagge, J. H., L. M. Tallaksen, L. Gudmundsson, A. F. V. Loon, and K. Stahl, 2015: Candidate Distributions for Climatological Drought Indices (SPI and SPEI). *International Journal of Climatology*, **35**, 4027–4040, <https://doi.org/10.1002/joc.4267>.
- Tomas-Burguera, M., S. M. Vicente-Serrano, D. Peña-Angulo, F. Domínguez-Castro, I. Noguera, and A. El Kenawy, 2020: Global Characterization of the Varying Responses of the Standardized Precipitation Evapotranspiration Index to Atmospheric

- Evaporative Demand. *Journal of Geophysical Research: Atmospheres*, **125**, e2020JD033017, <https://doi.org/10.1029/2020JD033017>.
- Van Loon, A. F., 2015: Hydrological drought explained. *Wiley Interdisciplinary Reviews: Water*, **2**, 359–392, <https://doi.org/10.1002/wat2.1085>.
- Vicente-Serrano, S. M., and S. Beguería, 2016: Comment on “Candidate distributions for climatological drought indices (SPI and SPEI)” by James H. Stagge et al. *International Journal of Climatology*, **36**, 2120–2131, <https://doi.org/10.1002/joc.4474>.
- Vicente-Serrano, S. M., and Coauthors, 2012: Performance of Drought Indices for Ecological, Agricultural, and Hydrological Applications. *Earth Interactions*, **16**, 1–27, <https://doi.org/10.1175/2012EI000434.1>.
- , D. Peña-Angulo, S. Beguería, F. Domínguez-Castro, M. Tomás-Burguera, I. Noguera, L. Gimeno-Sotelo, and A. El Kenawy, 2022: Global drought trends and future projections. *Philosophical Transactions of the Royal Society A: Mathematical, Physical and Engineering Sciences*, **380**, 20210285, <https://doi.org/10.1098/rsta.2021.0285>.
- WMO, 2012: *Standardized Precipitation Index*. http://www.wamis.org/agm/pubs/SPI/WMO_1090_EN.pdf (Accessed September 7, 2017).
- Worku, M. A., 2024: Spatiotemporal analysis of drought severity using SPI and SPEI: case study of semi-arid Borana area, southern Ethiopia. *Front. Environ. Sci.*, **12**, <https://doi.org/10.3389/fenvs.2024.1337190>.
- Wu, H., M. D. Svoboda, M. J. Hayes, D. A. Wilhite, and F. Wen, 2007: Appropriate application of the Standardized Precipitation Index in arid locations and dry seasons. *International Journal of Climatology*, **27**, 65–79, <https://doi.org/10.1002/joc.1371>.
- Yu, H., L. Wang, J. Zhang, and Y. Chen, 2023: A global drought-aridity index: The spatiotemporal standardized precipitation evapotranspiration index. *Ecological Indicators*, **153**, 110484, <https://doi.org/10.1016/j.ecolind.2023.110484>.

Reply to Referees

We sincerely appreciate the reviewers for dedicating their valuable time and providing insightful feedback that has significantly contributed to improving the paper.

Referee #1 (Remarks to the Author):

Thank you to the authors for such a thorough and attentive revision- I very much appreciate their hard work. I have only one small point I'd like to see the authors address before publication.

Authors' response:

- Thank you for acknowledging the improvements in the paper and for your constructive comments that helped us reach this level.

I continue to be baffled by the claim that global warming began to accelerate in 2000, and that this made the influence of AED more pronounced. This does not appear to be supported by anything in the analysis, the response to reviewers, or the literature. (Lenssen et al 2019 makes no statement about post-2000 acceleration, and to my knowledge none of the global temperature datasets show a step change in trends at this time). Moreover, I don't see that Figure 3a shows "the impact of AED has become increasingly evident since 2000", as claimed on line 257. What I see is a large trend in an hPET_Prc1m time series that passes through zero circa 2000 because that's the middle of the SPEI baseline period. This claim, which appears in the abstract and throughout the text, is a strong attribution statement and must be supported by strong evidence. My prior- subject to updating, of course- is that it's not true, and that high-frequency temporal variations on top of the long-term trends in global drought behavior are more readily explained by SST patterns, dynamics, and other factors. I suggest eliminating the "post-2000" claim and discussion of global warming acceleration or presenting quite a bit more evidence and rigorous attribution to justify it. It's fine to note the dramatic increase in drought are in the last five years- I think explaining the underlying drivers is probably out of scope here- and the discussion of the 2022 drought is now OK.

Authors' response:

- We appreciate the reviewer's comment on this matter. The text provided may have created some clarity issues, so we have removed all references to the post-2000 acceleration from the paper. As suggested, we have revised the abstract to focus on the increase in droughts over the last five years: "*During the last five years (2018–2022), the areas in drought have expanded by 74% on average compared to 1981–2017, with AED contributing to 58% of this increase*". We hope this revision effectively addresses the concern.

Referee #2 (Remarks to the Author):

I carefully read the new version of the manuscript and all the point-by-point replies by authors to the (I must admit) large number of suggestions made by the reviewers. I am positively surprised by the new version of the manuscript, the authors crafted robust and detailed responses and the paper is now ready, in my opinion for publication.

Authors' response:

- We thank the reviewer for carefully reading the manuscript. We are grateful for the suggestions and comments raised, which have greatly improved the paper.